# EngiAgent: Fully Connected Coordination of LLM Agents for Solving Open-ended Engineering Problems with Feasible Solutions

Xiyuan Zhou [* 1]   Ruixi Zou [* 2]   Xinlei Wang [* 3 4]   Yuheng Cheng [2]   Yan Xu [1]   Junhua Zhao [2 5]   Jinjin Gu [3]

## Abstract

Engineering problem solving is central to real-world decision-making, requiring mathematical formulations that not only represent complex problems but also produce feasible solutions under data and physical constraints. Unlike mathematical problem solving, which operates on predefined formulations, engineering tasks demand open-ended analysis, feasibility-driven modeling, and iterative refinement. Although large language models (LLMs) have shown strong capabilities in reasoning and code generation, they often fail to ensure feasibility, which limits their applicability to engineering problem solving. To address this challenge, we propose EngiAgent, a multi-agent system with a fully connected coordinator that simulates expert workflows through specialized agents for problem analysis, modeling, verification, solving, and solution evaluation. The fully connected coordinator enables flexible feedback routing, overcoming the rigidity of prior pipeline-based reflection methods and ensuring feasibility at every stage of the process. This design not only improves robustness to diverse failure cases such as data extraction errors, constraint inconsistencies, and solver failures, but also enhances the overall quality of problem solving. Empirical results across four representative domains demonstrate that EngiAgent achieves substantial improvements in feasibility compared to prior approaches, establishing a new paradigm for feasibility-oriented engineering problem solving with LLMs. Our source code and data are available at https://github.com/AI4Engi/EngiAgent.

*Equal contribution [1]Nanyang Technological University [2]The Chinese University of Hong Kong, Shenzhen [3]INSAIT, Sofia University "St. Kliment Ohridski" [4]The University of Sydney [5]AIRS. Correspondence to: Yan Xu <xuyan@ntu.edu.sg>, Junhua Zhao <zhaojunhua@cuhk.edu.cn>, Jinjin Gu <jinjin.gu@insait.ai>.

*Proceedings of the 43$^{rd}$ International Conference on Machine Learning*, Seoul, South Korea. PMLR 306, 2026. Copyright 2026 by the author(s).

## 1. Introduction

Large language models (LLMs) have shown strong capabilities in mathematical reasoning (Cobbe et al., 2021; Ahmaditeshnizi et al., 2024; Huang et al., 2025) and code generation (Wang et al., 2024; Dong et al., 2024; Wang et al., 2025), which has raised expectations that they could solve real-world engineering problems. However, engineering problems, such as designing transportation schedules or coordinating autonomous systems, are different from mathematical or coding tasks. They require not only correct formulations but also feasible solutions that must work under physical, operational, and safety constraints. In engineering, feasibility takes priority. Yet this requirement is largely ignored in current LLM research, creating a gap between recent progress and practical engineering demands.

Although prior work has explored diverse areas, feasibility has rarely been prioritized, limiting applicability to engineering problems. Research on autonomous research has considered open-ended exploration (Yamada et al., 2025; Schmidgall et al., 2025; Lu et al., 2024), but the focus is usually on generating novel ideas rather than producing executable and logically consistent solutions. As a result, our empirical analysis shows that fewer than 10% of the solutions are feasible. Research in mathematical modeling often stresses formulation quality over feasibility (Liu et al., 2025b; Astorga et al., 2025), with limited attention to generating feasible numerical solutions. A state-of-the-art (SOTA) model has been reported to achieve nearly 70% success on mathematical benchmarks, yet in our evaluation it generates numerical solutions for only about 13% of engineering problems. Research on code generation are effective in generating executable programs (Guo et al., 2024; Gao et al., 2023) but often neglects structured data requirements and physical constraints. Our empirical results show that while 62.26% of problems generate numerical outputs, only 5.66% are feasible. Overall, prior work has advanced open-ended exploration, mathematical formulation, and code generation, but consistently neglects the core requirement of producing feasible solutions in engineering.

Ensuring feasibility is challenging since engineering problems require solutions based on given data and constrained by physical and operational realities. Beyond producing cor-

rect formulations or executable code, solutions also need to satisfy physical laws, safety requirements, operational rules, and domain-specific data conditions. Feasibility may break down at multiple stages, including information extraction, constraint specification, and solver execution, where even minor errors can render solutions infeasible. In practice, solving complex problems often relies on agent-based systems. However, existing designs, whether fixed pipelines or single-agent setups, lack the flexibility to coordinate across stages and correct diverse sources of error.

To overcome these limitations, we propose EngiAgent, a framework that integrates multi-agent systems with a fully connected coordinator. The framework simulates expert practice by dividing the workflow into roles for problem analysis, modeling, verification, solving, and solution evaluation. These agents collaborate to ensure feasibility at each stage while maintaining modeling quality. Unlike prior methods with a fixed pipeline, the fully connected coordinator (Figure 1) dynamically routes feedback among agents, enabling targeted corrections when feasibility issues arise in data processing, constraint handling, or solver execution. This flexible structure enhances robustness and supports generalization across diverse engineering scenarios.

Empirical results demonstrate that EngiAgent achieves SOTA performance across three leading LLMs. The feasibility rates reach 64.15% on GPT-4o, 50.94% on Gemini-2.5 Flash, and 75.47% on DeepSeek-V3-671B, representing an average seven-fold improvement over prior SOTA methods. Furthermore, our fully connected coordinator improves feasibility by more than 10% on average compared with the fixed-pipeline framework.

Our contributions can be summarized as follows: (1) Feasibility is identified as a critical requirement for engineering modeling, and we collect a dataset covering four engineering domains with 53 high-quality problems to evaluate this aspect. (2) EngiAgent is proposed as the first framework designed to solve open-ended engineering problems by producing feasible solutions through a multi-agent system with a fully connected coordinator. (3) Empirical results demonstrate significant improvements in feasibility compared with previous approaches across diverse engineering tasks.

## 2. Related Works

**LLM Agents for Complex Problem.** With the advancement of LLMs, agents that combine autonomous reasoning, multi-step planning, and tool interaction have emerged as a new paradigm for complex problem solving (Huang et al., 2024b). Existing approaches, however, are typically built on fixed pipelines with predetermined workflows. They can be grouped into three categories: sequential planning agents that execute step-by-step processes (Yao et al., 2023; Yang

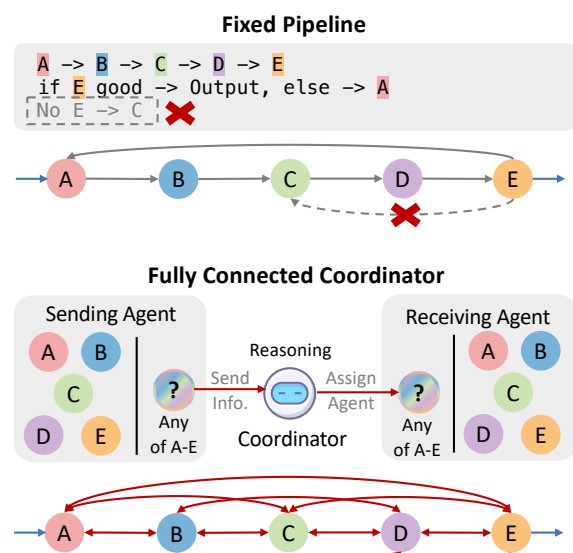

*Figure 1.* Comparison of fixed pipeline and fully connected coordinator. Fixed pipelines enforce rigid execution, whereas the fully connected coordinator enables dynamic routing and flexible error handling across agents.

et al., 2023), hierarchical task decomposition methods (Shen et al., 2023; Liang et al., 2024), and multi-agent coordination frameworks with fixed roles and protocols (Chen et al., 2023; Pan et al., 2025; Grötschla et al., 2025). These systems rely on static routing and lack adaptive error handling across formulation, modeling, verification, and solving.

Such rigidity becomes a bottleneck in open-ended engineering problems, which feature multi-objective trade-offs, evolving requirements, and interdependent constraints. Recent studies report systematic failure modes in multi-agent systems, including poor design, mis-specified tasks, inter-agent misalignment, and verification deficiencies (Cemri et al., 2025). Unlike structured computational tasks with clear success metrics, engineering challenges demand adaptive formulation, dynamic constraint handling, and trade-off navigation under real-world feasibility requirements. This gap underscores the need for flexible coordination mechanisms that can dynamically adapt, debug, and refine workflows for engineering problem solving.

**LLMs for Engineering.** Recently, advances in LLM reasoning have expanded applications in engineering. With cross-domain knowledge, multi-step reasoning, and planning abilities, LLMs provide new pathways for modeling and solving complex engineering tasks. However, most studies remain limited to closed tasks and specific scenarios, lacking general modeling and iterative solving capabilities for open-ended engineering problems. Existing work has mainly focused on power systems (Yan & Xu, 2023; Cheng et al., 2025; Majumder et al., 2024), materials (Liu et al.,

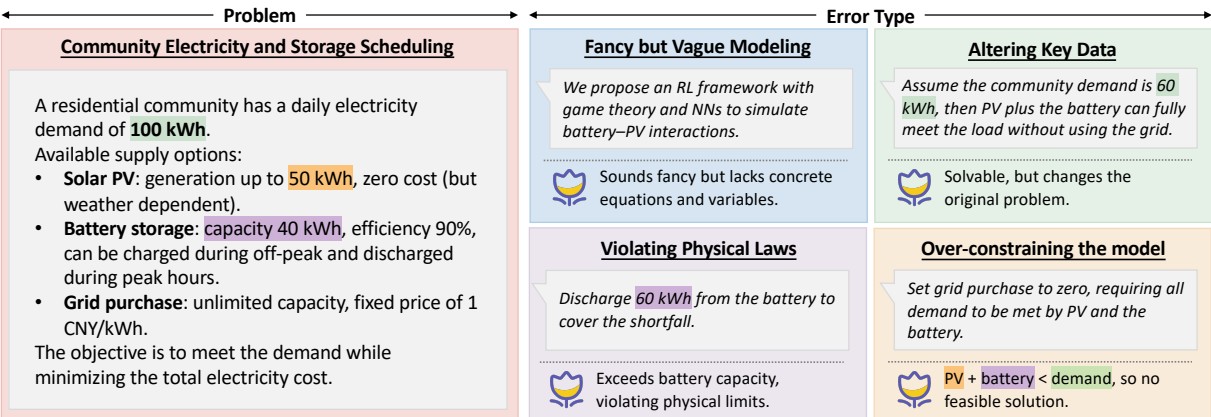

*Figure 2.* Illustration of common error types in engineering problem solving with a community electricity and storage scheduling e: fancy but vague modeling, altering key data, violating physical laws, and over-constraining the model.

2024; Jiang et al., 2025; Buehler, 2024), structures (Antunes et al., 2024; Jiang et al., 2023), mechanical design (Buehler, 2024; Xiaorui et al., 2025; Ni & Buehler, 2024), manufacturing (Zhou et al., 2024; Xia et al., 2024), and transportation (Yang et al., 2024; Qu et al., 2023), typically using fixed templates or tool-calling pipelines. While effective in predefined contexts, these approaches struggle with open problems involving uncertain objectives, complex constraints, and dynamic feedback, and fail to build general systems for cross-task modeling, solving, and optimization.

Related research is autonomous research, which seeks to automate the scientific workflow from problem generation and literature review to experiment execution and paper writing. Current studies fall into two directions: multi-stage language generation and planning for general scientific tasks (Yamada et al., 2025; Schmidgall et al., 2025; Lu et al., 2024; Huang et al., 2024a; Liu et al., 2025a; Baek et al., 2025), and domain-specific automation in areas such as biomedical discovery (Gao et al., 2024), chemistry (Darvish et al., 2025), and transportation analysis (Guo et al., 2025). Compared with autonomous research, which emphasizes generating scientifically meaningful ideas and problem formulations, open-ended engineering automation focuses on structuring real-world problems, integrating professional tools and physical constraints, and producing feasible solutions through iterative feedback. This requires structured modeling, optimized tool use, feasibility verification, and adaptive refinement, extending autonomous research toward practical engineering problem solving.

## 3. Feasibility Challenges in Engineering Problem Solving

In engineering problems, feasibility must take priority. The goal is not only to derive a formally correct model or numerical result, but to obtain a solution that can be implemented under real-world conditions. A solution that is logically consistent and mathematically valid becomes meaningless if it violates data consistency, physical laws, operational safety, or procedural constraints. Thus, the value of an engineering solution depends primarily on its feasibility.

In this work, a solution is considered feasible only if it satisfies all mandatory engineering constraints required for execution in real systems, such as data consistency, physical requirements, safety limits, and procedural rules.

This stands in contrast to mathematical problems, where correctness directly implies feasibility. Engineering problems are inherently open-ended, allowing multiple modeling choices. For example, simple versus complex formulations or linear versus nonlinear optimization. While many alternatives may appear logically sound, only a small fraction yield solutions that remain feasible under real-world constraints. This difference makes feasibility particularly difficult to ensure in engineering problem solving, since logical correctness alone does not guarantee implementability.

As shown in Figure 2, common error types in engineering problem solving can be categorized into four groups:

1. Fancy but Vague Modeling: Methods that employ sophisticated terminology or advanced paradigms but fail to specify explicit variables, constraints, or equations. Such formulations sound appealing but cannot be transformed into computable or executable solutions.

2. Altering Key Data: Approaches that arbitrarily modify or replace critical data or conditions from the original problem. While this may lead to solvable models, the results no longer correspond to the intended problem and thus lack practical validity.

3. Violating Physical Laws: Models that ignore fundamental physical principles or engineering constraints,

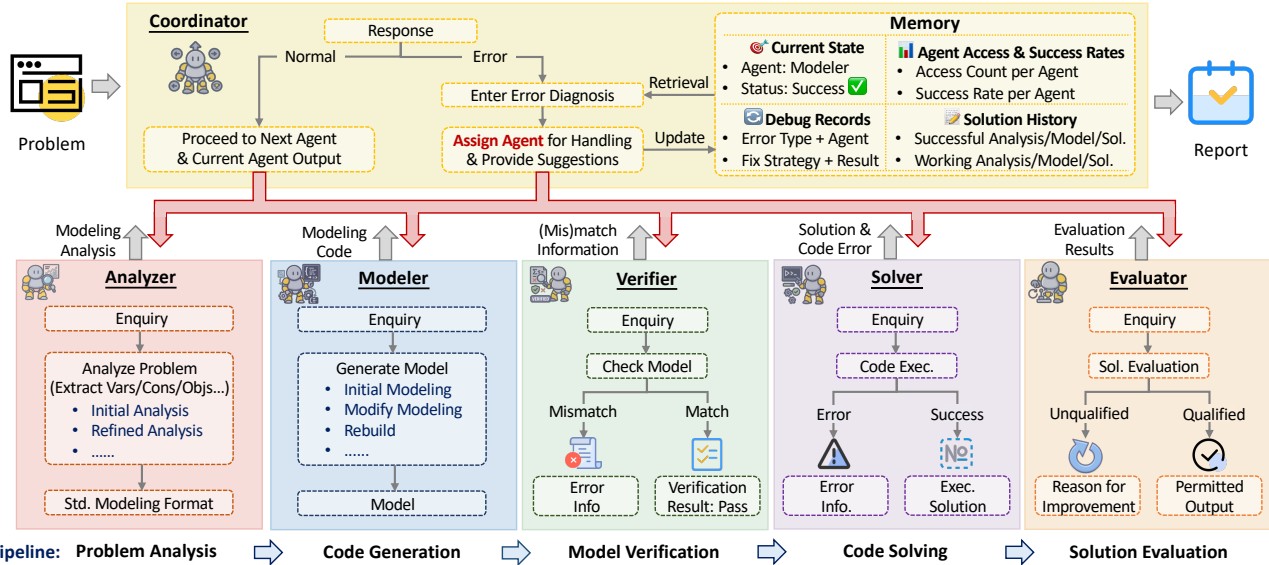

*Figure 3.* Overall Workflow of EngiAgent. The system consists of five functional agents (Analyzer, Modeler, Verifier, Solver, Evaluator) together with a Fully Connected Coordinator and shared Memory. The baseline pipeline (blue arrows) shows the predefined sequential cognitive architecture, while the coordinator layer (red arrows) enables flexible routing, error diagnosis, and adaptive scheduling across agents, ensuring feasibility and efficiency in solving open-ended engineering problems.

producing solutions that may be mathematically valid but physically impossible, such as exceeding capacity, efficiency, or safety limits.

4. Over-constraining the Model: Solutions that introduce excessive or unrealistic constraints. Although the formulation may appear more rigorous, the additional restrictions can eliminate all feasible solutions under real-world conditions.

Thus, although existing methods have made progress in areas such as open-ended exploration, mathematical modeling, and code generation, they still struggle to ensure feasibility in engineering contexts. A key reason is that these approaches often sacrifice feasibility in pursuit of other objectives. For instance, open-ended exploration emphasizes novelty, which frequently results in *fancy but vague modeling*. Mathematical modeling tends to focus on proposing elegant frameworks rather than executable solutions, which can lead both to *vague formulations* and to *over-constraining the model*. Code generation methods, while capable of producing runnable programs, often achieve this by altering data or modifying constraints, thereby introducing errors such as *altering key data* or *violating physical laws*.

These observations highlight a limitation of current research: they advance correctness and expressiveness but fail to guarantee feasibility, which is the essence of engineering problem solving. Addressing this gap requires new frameworks that explicitly prioritize feasibility at every stage of the modeling and solving process, ensuring solutions are both

logically valid and practically applicable.

# 4. EngiAgent

**Overview.** EngiAgent is a multi-agent system for solving open-ended engineering problems with feasible solutions. *We structure the framework following the standard engineering problem-solving workflow by coordinating five specialized agents: Analyzer, Modeler, Verifier, Solver, and Evaluator.* This rigorous organization enables systematic execution of problem analysis, code generation, model verification, code solving, and solution evaluation. Prior frameworks typically adopt fixed pipelines, limiting adaptability. In contrast, EngiAgent introduces a Fully Connected Coordinator that orchestrates interactions and leverages Memory to support error diagnosis, state-aware reasoning, and structured feedback routing. This design moves beyond rigid pipelines and enables robust and efficient coordination across the end-to-end workflow (Figure 3).

**Fully Connected Coordinator.** The Fully Connected Coordinator serves as the control center of the automated modeling and solving workflow. The coordinator employs a hybrid control strategy that grounds LLM-based autonomous decision-making within a structured engineering framework. Rather than following a rigid set of instructions, this mechanism provides engineering protocols as contextual guidelines, enabling the LLM to dynamically determine the effective routing path based on real-time feedback. To address the potential instability of open-ended reasoning, the system incorporates state-aware memory and a forced agent switch-

ing mechanism. By tracking error history and triggering agent transitions when repetitive failures occur, this design effectively constrains divergent behavior, promoting system convergence and preventing infinite debugging loops. Through this guideline-driven reasoning, the coordinator identifies the root causes of failures, directs specific repair tasks to the most appropriate agents, and makes reliable termination decisions based on comprehensive evaluation feedback. This optimizes the collaboration process to correctly and efficiently generate engineering-feasible solutions. The contribution of the Fully Connected Coordinator and the forced switching mechanism is validated by the ablation study in Section 6.5.

**Analyzer.** The Analyzer serves as the starting point of the modeling process, converting natural language descriptions of engineering problems into structured problem formulations through LLM-based semantic parsing. It processes raw problem statements alongside knowledge of library retrieval to extract decision variables, parameters, constraints, and objectives. In addition to extracting explicit information, the Analyzer identifies implicit engineering rules and physical constraints that may not be explicitly expressed in the text, and further addresses uncertainty, trade-offs and sensitivity analysis. Dealing with retry requests, the agent incorporates mismatch information from verification and targeted modification strategies under coordinator guidance, and it supports reanalysis in response to code errors by adapting the modeling approach. Its output contains modeling context, core elements, and extended analysis, providing a reliable foundation for subsequent stages.

**Modeler.** The Modeler is the core modeling module that converts structured problem formulations into executable models via template-based code generation. Leveraging domain-specific modeling knowledge, it reduces common errors in engineering modeling and ensures correct handling of specifications, data inputs, and variable usage. When retries are triggered, the Modeler applies targeted revisions based on coordinator guidance, focusing on modeling expression errors and mismatches between problem types and solution formats. It also performs consistency checks to reduce implementation errors such as index mismatches, improper constraint formatting, and incorrect variable definitions. In addition, the Modeler supports efficient modeling of large-scale problems while preserving mathematical rigor and computational feasibility.

**Verifier.** The Verifier functions as a strict checking module that ensures the generated model is consistent with the original engineering problem. It examines the model from three aspects: semantic consistency, constraint completeness, and data consistency. To maintain correctness without reducing system efficiency, the Verifier adopts a hierarchical decision process. First, it strictly checks non-negotiable semantic

correctness: any critical error, such as an incorrect objective direction, missing core physical laws, or inconsistent data, leads to immediate rejection. Second, it distinguishes substantive logical errors from minor implementation differences by recognizing functionally equivalent formulations that vary only in representation, thereby avoiding repeated debugging loops caused by formatting differences. At the same time, the Verifier explicitly disallows arbitrary deletion or modification of fundamental hard constraints. The Verifier produces detailed error diagnostics with explicit reasons, enabling the Coordinator to route targeted feedback to the most appropriate agent for correction. This reduces unnecessary iterations and downstream computation, and ensures that the final model is logically sound and compliant with engineering requirements. The Verifier's contribution is validated by the ablation study in Section 6.5.

**Solver.** The Solver executes the generated models and manages the solving process. It selects appropriate solver backends for each task and enforces time and resource limits to avoid deadlocks and excessive computation. The Solver distinguishes valid results from execution errors and returns informative feedback for debugging and recovery. This design ensures reliable solving while maintaining system stability and efficiency.

**Evaluator.** The Evaluator performs overall quality assessment of complete engineering solutions and provides improvement suggestions while balancing technical rigor and practical feasibility. It evaluates solution packages that contain problem descriptions, modeling analyses, implementation code, and computational results along four dimensions: result feasibility, model problem alignment, engineering validity, and overall solution quality. Using LLM-based assessment, it conducts multi-perspective analysis and scoring, generates evaluation reports, and determines whether to restart the process or output the final solution. This holistic evaluation ensures that the final solution meets practical standards and identifies key areas for further refinement.

## 5. Construction of the Feasible Solution Validation

**Problem Collection.** We construct a benchmark to explicitly evaluate solution feasibility in engineering problem solving. The benchmark covers four domains: (1) market and multi-agent decision-making, (2) scheduling and resource allocation, (3) planning and design, and (4) control and autonomous system modeling. Problems are collected from high-quality peer-reviewed papers and rewritten into self-contained open-ended modeling tasks with background context, numerical data, and explicit instructions. Each task is further equipped with explicit engineering constraints derived from both the original papers and real-world requirements. These constraints define the conditions that

*Table 1.* Experimental results on open-ended engineering solving. Total (all problems) and Feasible (subset of feasible solutions) are reported. The Feas. column (highlighted in gray) is the key metric, indicating the ability to generate solutions that satisfy real-world constraints. EngiAgent (Coord.) is our proposed system with a fully connected coordinator. Best results are in red, second best in blue. Abbreviations: Num. = numerical solution rate, Feas. = feasible solution rate, IE = Information Extraction, DR = Domain-specific Reasoning, MO = Multi-objective Decision-making, UH = Uncertainty Handling, Avg. = Average score.

| Methods | Num. ↑ | Feas. ↑ | Total | | | | | Feasible | | | | |
| | | | IE ↑ | DR ↑ | MO ↑ | UH ↑ | Avg. ↑ | IE ↑ | DR ↑ | MO ↑ | UH ↑ | Avg. ↑ |
|---|---|---|---|---|---|---|---|---|---|---|---|---|
| **GPT-4o** | | | | | | | | | | | | |
| Zero-shot | 22.64% | 5.66% | 5.66 | 5.42 | 4.47 | 3.33 | 4.72 | 5.67 | 5.10 | 1.33 | 4.00 | 4.03 |
| ResearchAgent | 15.09% | 3.77% | 5.36 | 5.25 | 5.10 | 5.17 | 5.22 | 5.50 | 5.50 | 6.00 | 5.50 | 5.63 |
| DS-Agent | **62.26%** | 5.66% | 5.91 | 4.83 | 4.85 | 4.79 | 5.10 | 6.00 | 5.75 | 5.38 | 6.25 | 5.85 |
| MM-Agent | 13.21% | 7.55% | 6.89 | 7.21 | 6.48 | **7.98** | 7.14 | 8.25 | **8.38** | **7.88** | 7.50 | **8.00** |
| EngiAgent (Fixed) | 47.17% | **47.17%** | **8.30** | **7.22** | **6.67** | 7.14 | **7.33** | **8.84** | 8.12 | 7.40 | **7.56** | 7.98 |
| EngiAgent (Coord.) | **66.04%** | **64.15%** | **8.67** | **7.74** | **7.05** | **7.41** | **7.72** | **8.86** | **8.13** | **7.53** | **7.94** | **8.12** |
| **Gemini-2.5 Flash** | | | | | | | | | | | | |
| Zero-shot | 0.00% | 0.00% | 6.80 | 5.67 | 5.53 | 3.55 | 5.39 | / | / | / | / | / |
| ResearchAgent | 1.89% | 0.00% | 4.56 | 4.41 | 4.34 | 4.36 | 4.42 | / | / | / | / | / |
| DS-Agent | **60.38%** | 0.00% | 7.17 | **6.75** | **6.71** | 5.85 | **6.62** | / | / | / | / | / |
| MM-Agent | 9.43% | 3.77% | 6.39 | 5.72 | 5.98 | **6.40** | 6.12 | 8.00 | **8.00** | **8.25** | **6.00** | **7.56** |
| EngiAgent (Fixed) | 45.28% | **39.62%** | **7.30** | **6.93** | 5.82 | 5.21 | 6.32 | **8.60** | 6.71 | 6.14 | 5.36 | 6.70 |
| EngiAgent (Coord.) | **52.83%** | **50.94%** | **8.30** | 6.89 | **6.30** | **6.06** | **6.89** | **8.80** | **7.23** | **6.50** | **6.81** | **7.34** |
| **DeepSeek-V3-671B** | | | | | | | | | | | | |
| Zero-shot | 0.00% | 0.00% | 6.20 | 5.60 | 5.09 | 3.99 | 5.22 | 6.50 | 6.25 | 6.50 | 6.50 | 6.44 |
| ResearchAgent | 9.43% | 7.55% | 5.12 | 4.89 | 4.84 | 4.98 | 4.96 | 6.50 | 5.98 | 4.92 | 4.98 | 5.60 |
| DS-Agent | **77.36%** | 28.30% | 7.39 | 6.88 | 6.77 | 6.01 | 6.76 | 8.22 | 7.64 | 7.38 | 7.12 | 7.59 |
| MM-Agent | 11.32% | 5.66% | **8.98** | **8.84** | **7.64** | **7.01** | **8.12** | **9.00** | **8.50** | **8.00** | **7.85** | **8.34** |
| EngiAgent (Fixed) | 73.58% | **67.92%** | **8.08** | 7.28 | **7.20** | **6.98** | **7.39** | **8.38** | 7.45 | **7.53** | **7.33** | **7.67** |
| EngiAgent (Coord.) | **79.25%** | **75.47%** | 7.85 | **7.42** | 7.08 | 6.59 | 7.24 | 8.05 | **8.16** | 7.26 | 6.52 | 7.50 |

must be satisfied for a solution to be considered feasible under realistic scenarios. Following the design principles established in EngiBench (Zhou et al., 2025), we adopt four dimensions to assess modeling and reasoning quality: information extraction, domain-specific reasoning, multi-objective decision-making, and uncertainty handling. These dimensions provide a systematic evaluation framework for engineering problem solving. To guarantee quality and relevance, more than 20 experienced domain experts reviewed and refined all tasks. Details are provided in Appendix B.

**Task Formation.** Each task requires the model to extract key information from the text, including variables, parameters, constraints, and objectives, and then construct a numerical model to produce concrete solutions. The tasks are designed to preserve openness while still guiding the process toward verifiable numerical solving, which enables a faithful evaluation of the modeling and reasoning capabilities of LLMs in engineering problems.

**Evaluation.** Model performance is first evaluated by feasibility, which is the most critical dimension. Feasibility determines whether the final solution is consistent with the core data provided in the problem statement and whether

it satisfies all specified constraints. It is assessed as a binary variable, where 0 indicates infeasible and 1 indicates feasible, based on executable numerical and constraint verification with final confirmation by human experts rather than any LLM-based scoring. Beyond feasibility, the quality of modeling and reasoning is evaluated along four dimensions, quantitatively scored on a scale from 0 to 10.

## 6. Experiments

### 6.1. Baselines.

We evaluate EngiAgent against SOTA LLM agents for problem solving. Since no prior work explicitly targets open-ended engineering problems with feasibility requirements, we adapt existing LLM-based agents for comparison. The baselines include: (1) Zero-shot prompting, where the LLM directly attempts to solve problems without structured guidance; (2) ResearchAgent (Huang et al., 2024a), originally designed to automate experimentation loops in machine learning, which we extend to engineering tasks; (3) DS-Agent (Guo et al., 2024), an agent framework for automated data science based on case-based reasoning, adapted here

*Table 2.* Experimental results on runtime, token usage, and inference cost of different agents across three LLMs. Duration is measured in seconds, and tokens represent the total number of input and output tokens processed. Cost is estimated in USD based on the official pricing of each model: GPT-4o at $2.50 / 1M input tokens and $5.00 / 1M output tokens, Gemini-2.5 Flash at $0.30 / 1M input tokens and $2.50 / 1M output tokens, and DeepSeek-V3-671B at $0.27 / 1M input tokens and $1.10 / 1M output tokens.

| Methods | GPT-4o | | | Gemini-2.5 Flash | | | DeepSeek-V3-671B | | |
|---|---|---|---|---|---|---|---|---|---|
| | Duration (s) | Tokens | Cost ($) | Duration (s) | Tokens | Cost ($) | Duration (s) | Tokens | Cost ($) |
| Zero-shot | 112 | 6,986 | 0.02 | 46 | 20,767 | 0.04 | 110 | 7,893 | 0.01 |
| ResearchAgent | 306 | 181,324 | 0.63 | 85 | 37,134 | 0.04 | 218 | 14,943 | 0.01 |
| DS-Agent | 243 | 41,688 | 0.13 | 455 | 196,014 | 0.22 | 786 | 75,337 | 0.04 |
| MM-Agent | 1231 | 280,452 | 0.81 | 1714 | 648,042 | 0.44 | 1643 | 263,113 | 0.11 |
| EngiAgent (Fixed) | 629 | 154,352 | 0.47 | 1743 | 867,297 | 0.90 | 1650 | 198,657 | 0.09 |
| EngiAgent (Coord.) | 659 | 171,941 | 0.51 | 1119 | 813,157 | 0.67 | 1251 | 178,077 | 0.07 |

to engineering modeling and solving; and (4) MM-Agent (Liu et al., 2025b), a framework for mathematical modeling tasks, repurposed for engineering problems by evaluating its ability to construct models and generate solutions.

## 6.2. Experimental Results

**Limitations of existing methods.** Experiments (Table 1) show that existing frameworks achieve very low feasibility rates on engineering tasks. Each method has partial strengths: DS-Agent attains a high numerical solution rate, but most outputs either mismatch the problem data or violate constraints, making them infeasible. MM-Agent can generate more feasible solutions, yet its overall ability to produce numerical results is too limited, often remaining at the level of textual modeling. Concretely, the best baseline feasible rates are only 7.55% on GPT-4o, 3.77% on Gemini-2.5 Flash, and 28.30% on DeepSeek-V3-671B.

**Overall advantage of EngiAgent.** On all three tested LLMs, EngiAgent (Coord.) consistently surpasses the strongest prior baselines in generating feasible solutions. On GPT-4o, the feasibility rate increases from 7.55% for the best baseline to 64.15% with EngiAgent. On Gemini-2.5 Flash, it rises from 3.77% to 50.94%. On DeepSeek-V3-671B, it increases from 28.30% to 75.47%. These results clearly highlight EngiAgent's substantial advantage in delivering feasible engineering solutions.

**EngiAgent reduces the gap between numerical and feasible solutions.** Experiments show that EngiAgent greatly improves the consistency between numerical outputs and feasible solutions, ensuring that most generated results are valid. By contrast, existing methods often produce large gaps: for example, DS-Agent attains 62.26% numerical solutions on GPT-4o but only 5.66% are feasible; on DeepSeek-V3-671B it achieves 77.36% numerical solutions but just 28.30% feasible. MM-Agent shows smaller gaps but generates too few numerical results overall. EngiAgent (Coord.), however, maintains near consistency across all LLMs, with gaps as small as 1.89% on GPT-4o, 1.89% on Gemini-2.5 Flash, and

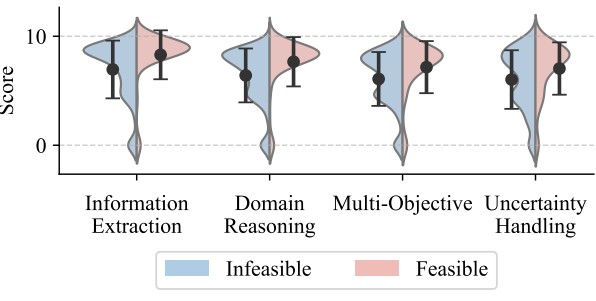

*Figure 4.* Distribution of evaluation scores across four dimensions for feasible and infeasible solutions. Scores above 10 are artifacts of kernel density estimation (KDE) smoothing.

3.78% on DeepSeek-V3-671B, indicating that most numerical solutions satisfy real-world constraints. Representative output examples are provided in Appendix D.

**Contribution of the fully connected coordinator.** Ablation studies confirm the importance of the fully connected coordinator. Replacing it with a fixed pipeline EngiAgent (Fixed) causes consistent drops in both the numerical and feasible rates, with average declines of exceeding 10% across the three models. This shows that flexible coordination is essential for robust feasibility in engineering tasks.

## 6.3. Limitations of Purely Text-Based Judgments

In recent studies, text-based judgment has become a common approach to evaluating LLM performance on open-ended tasks. These methods, often powered by rubric-based scoring or LLM-as-a-judge frameworks, assign scores based on the linguistic quality, logical structure, and surface plausibility of model outputs. While convenient, such judgments are poorly aligned with the requirements of engineering problem solving, where feasibility is paramount.

As illustrated in Table 1 and Figure 4, text-based scores often appear inflated. Many solutions that are infeasible under engineering constraints nevertheless receive relatively high ratings because they are well-written or logically coherent

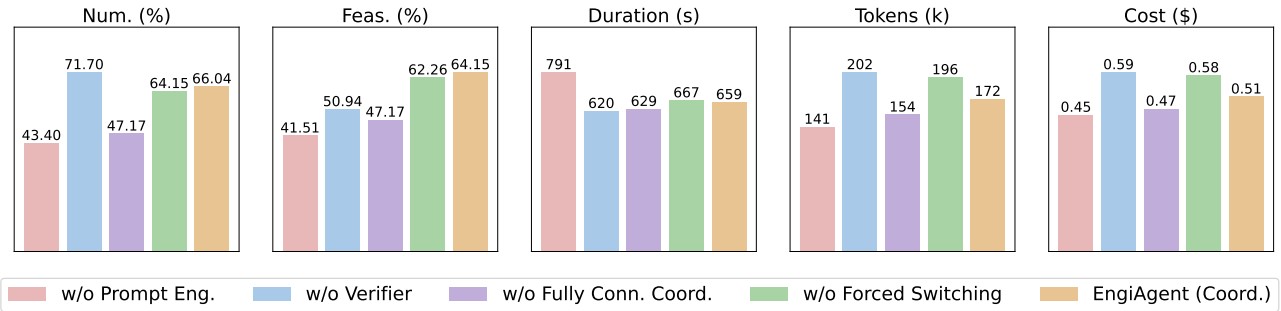

*Figure 5.* Ablation results over five metrics comparing degraded variants and the full EngiAgent.

at the textual level. This creates a misleading distribution in which feasible and infeasible solutions substantially overlap, obscuring the true capability gap. For example, solutions that violate energy conservation, exceed device capacity, or ignore operational safety limits may still obtain high textual scores, despite being unusable in practice.

This discrepancy reveals a limitation: purely text-based judgments reward expression rather than execution. They assess whether an answer looks convincing but cannot verify whether it actually satisfies numerical requirements, physical laws, or real-world operational constraints. Consequently, relying solely on such evaluations risks overestimating model performance in engineering domains. To address this issue, feasibility-based validation must be incorporated, ensuring that evaluation reflects not only linguistic plausibility but also practical implementability.

### 6.4. Cost Efficiency Analysis

We further evaluate the runtime efficiency of different agents across three leading LLMs (Table 2), considering duration, token usage, and cost. Zero-shot prompting is the cheapest but has almost no feasible solutions, offering little practical value. Baseline agents such as DS-Agent and ResearchAgent incur relatively low costs but perform poorly in engineering feasibility. MM-Agent achieves stronger results in some dimensions but suffers from substantially higher inference overhead, with both runtime and token consumption at the highest level, resulting in poor overall cost-effectiveness.

In contrast, EngiAgent (Coord.) demonstrates consistently strong cost efficiency across all three LLMs. While its runtime and token usage are higher than those of some baseline agents, it achieves markedly higher feasibility rates while keeping inference costs within a reasonable range. The average cost on GPT-4o is $0.51, Gemini-2.5 Flash is $0.67, and on DeepSeek-V3-671B it is as low as $0.07, which is substantially lower than that of traditional multi-round frameworks. These results highlight that EngiAgent achieves a favorable balance between feasibility and efficiency.

A direct comparison with the baseline variant EngiAgent

(Fixed) highlights the effectiveness of the fully connected coordinator. While overall costs remain similar across all three LLMs, EngiAgent (Coord.) consistently achieves higher feasibility. On GPT-4o, feasibility increases by +16.98% at nearly the same cost ($0.51 vs. $0.47). On Gemini-2.5 Flash, it reduces both cost ($0.67 vs. $0.90) and tokens (813k vs. 867k) while improving feasibility by +11.32%. On DeepSeek-V3-671B, it further lowers cost ($0.07 vs. $0.09) and runtime (1251s vs. 1650s), with feasibility gains exceeding +7.55%. These results confirm that the fully connected coordinator substantially improves robustness and feasibility without increasing overall computation.

### 6.5. Ablation Study

To investigate the contribution of each module in EngiAgent, we conduct an ablation study on GPT-4o. Specifically, we analyze the effects of prompt engineering, the verifier, the fully connected coordinator, and the forced agent switching mechanism. As shown in Figure 5, we evaluate four degraded variants: (1) removing prompt engineering while keeping only minimal task instructions (w/o prompt eng.); (2) removing the verifier and relying solely on the remaining agents for modeling and solving (w/o verifier); (3) replacing the fully connected coordinator with a fixed sequential workflow (w/o fully connected coordinator, i.e., EngiAgent (Fixed)); and (4) disabling forced agent switching so that error correction relies only on the coordinator's internal reasoning (w/o forced switching).

Experimental results show that prompt engineering is important for accurate information extraction and modeling guidance, and its removal significantly reduces both the numerical-solution rate and feasibility. Removing the verifier leads to data inconsistencies and constraint omissions, causing a large drop in feasibility and reproducing typical failure types discussed in Section 3. In addition, EngiAgent (Fixed) lacks the ability to dynamically route errors to appropriate agents, resulting in simultaneous degradation in both numerical outputs and feasibility, consistent with the observations reported in Section 6.2. Forced agent switching only serves as a last-resort mechanism triggered when repetitive

**Problem**

---

**Market-Based Power Allocation with Multiple Supply and Demand Units**

An exporting province $S$ supplies two receiving provinces $R1$ and $R2$ through uncertain DC corridors, with an AC interconnection between $R1$ and $R2$. The objective is to maximize expected social welfare while reducing network stress under two scenarios (High 0.6, Low 0.4).

- **Supply (S):** $S1$: 120 @ 20, $S2$: 80 @ 24, $S3$: 60 @ 28
- **Demand:** L1: 100 @ 70, L2: 80 @ 60, M1: 120 @ 65, M2: 90 @ 55
- **Local Supply:** G1: 90 @ 30, G2: 60 @ 35, H1: 70 @ 32, H2: 50 @ 40

**Report Output**

---

**Pyomo Code Snippet**

```
# === Parameters from problem statement ===
supply_cost = {'S1': 20, 'S2': 24, 'S3': 28,
        'G1': 30, 'G2': 35, 'H1': 32, 'H2': 40}
demand_value = {'L1': 70, 'L2': 60, 'M1': 65, 'M2': 55}
# === Objective: weighted welfare - stress ===
model.obj = Objective(
    expr = gamma * welfare - (1 - gamma) * stress,
    sense = maximize)
```

**Optimal Results ($\gamma = 0.7$)**

**High Scenario (H)**
- $x_{S1} = 120$, $x_{S2} = 80$, $x_{S3} = 60$
- $f_{DC1} = 10$, $f_{DC2} = 250$, $f_{AC} = 80.7$
- $y_{L1} = 100$, $y_{L2} = 80$, $y_{M1} = 120$, $y_{M2} = 90$

**Low Scenario (L)**
- $f_{DC1} = 80$, $f_{DC2} = 180$, $f_{AC} = 15.6$
- Other allocations remain the same as in the high scenario, satisfying balance and constraints.

*Figure 6.* An example output generated by EngiAgent. The problem statement is extracted from a power system market allocation task with multiple supply and demand units and transformed into a modeling input. The system automatically produces the corresponding Pyomo code snippet and solves it under given parameters to obtain the optimal results. The outputs report key power flow allocations and local generation across high and low scenarios, confirming feasibility and constraint satisfaction.

debugging cycles emerge; disabling it does not noticeably affect solution quality, but slightly increases unnecessary retries and incurs additional computational overhead.

In summary, prompt engineering and the verifier provide essential modeling and feasibility checks, while the fully connected coordinator and forced switching enhance adaptive routing and debugging, enabling EngiAgent to produce feasible solutions in open-ended engineering tasks.

### 6.6. Example Output of EngiAgent

To demonstrate the characteristics of EngiAgent's outputs, we present an example based on problem statement data (Figure 6). All results are strictly grounded in the numerical parameters provided in the problem. The system first transforms the natural language description into a standardized modeling format, then the Modeler automatically generates Pyomo code. Optimization is performed within this code to produce explicit numerical results. The final output not only provides concrete values for key variables but also confirms feasibility and constraint satisfaction.

## 7. Conclusion

This paper addresses the core challenge that LLMs often fail to guarantee feasibility in engineering problem solving. We propose EngiAgent, a multi-agent framework equipped with a fully connected coordinator that enables flexible feedback and targeted correction across problem analysis, modeling, verification, solving, and evaluation. This design

substantially enhances robustness and ensures feasibility throughout the workflow. Experimental results demonstrate that EngiAgent significantly outperforms existing methods across multiple SOTA LLMs, achieving higher rates of feasible solutions and narrowing the gap between numerical outputs and practically valid results. Moreover, EngiAgent attains strong feasibility while maintaining reasonable efficiency and cost, highlighting its practicality and potential for broader application. Overall, this study establishes a new pathway toward feasibility-oriented intelligent systems for engineering applications.

## Impact Statement

This work introduces EngiAgent, a multi-agent framework for generating feasible solutions to open-ended engineering problems. By prioritizing constraint satisfaction and physical validity, the proposed approach aims to support more reliable automated modeling and decision-making in engineering domains. However, misuse or overreliance on automated solutions may lead to unsafe outcomes when problem specifications are incomplete or incorrect. To mitigate these risks, EngiAgent emphasizes feasibility verification and transparent intermediate outputs, enabling effective human oversight. We view this system as an assistive tool for engineers rather than a replacement for expert judgment, and encourage further research on safe and responsible deployment in real-world engineering settings.

## Acknowledgements

This work was supported in part by the Ministry of Education and Science of Bulgaria (support for INSAIT, part of the Bulgarian National Roadmap for Research Infrastructure), the Shenzhen Institute of Artificial Intelligence and Robotics for Society (AIRS), the Shenzhen Key Laboratory of Crowd Intelligence Empowered Low-Carbon Energy Network (No. ZDSYS20220606100601002), the National Natural Science Foundation of China (No. 72331009). Y. Xu's work was supported by the National Research Foundation, Singapore (NRF), Maritime and Port Authority of Singapore (MPA), and Singapore Maritime Institute (SMI) under its Maritime Transformation Programme (Project No. SMI-2024-MTP-01). Any opinions, findings, conclusions, or recommendations expressed in this material are those of the author(s) and do not reflect the views of NRF, MPA, or SMI.

We would like to express our sincere gratitude to Mo Chen, Jiaxiang Xie, Bihua Wen, Jili Tu, Zhuoqi Li, Kaicheng Li, Rui Jin, Zixuan Cui, Yuhao Wu, Yirui He, and Yulu Xie for their valuable contributions to the construction, annotation, and validation of the dataset used in EngiAgent.

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

# Appendix

## A. The Use of Large Language Models

In this work, LLMs were used for grammar checking and language polishing during paper writing, and within the method they were invoked as intelligent agents in the EngiAgent framework to perform reasoning and modeling tasks.

## B. Dataset for EngiAgent Evaluation

### B.1. Classification of Problem Domains

The first category is *Market and Multi-Agent Decision-Making*. This category emphasizes the strategic interactions among multiple autonomous participants, with a particular focus on equilibrium modeling and competitive bidding. Typical tasks include electricity market bidding strategies, carbon market equilibrium analysis, and price forecasting under uncertainty.

The second category is *Scheduling and Resource Allocation*. This category centers on the optimization of resources, tasks, and time within a unified control framework. Representative examples involve production scheduling, task assignment, routing optimization, and unmanned vehicle mission allocation.

The third category is *Planning and Design*. This category addresses the optimization of layouts and facility locations for complex systems. Problems in this area include power grid expansion, infrastructure deployment, structural optimization, and urban pipeline planning.

The final category is *Control and Autonomous System Modeling*. This category highlights dynamic modeling and real-time feedback control of autonomous systems. Example tasks include UAV swarm coordination, trajectory planning with obstacle avoidance, adaptive control under disturbances, and autonomous navigation.

### B.2. Category Statistics

To ensure comprehensive coverage of engineering challenges, the dataset is constructed with a balanced yet diverse distribution of problems. Table 3 summarizes the number of tasks in each category. In total, the dataset contains fifty-three tasks, distributed across four major categories.

*Table 3.* Statistics of problem categories in the dataset.

| Category | Number of Tasks |
|---|---|
| Market and Multi-Agent Decision-Making | 21 |
| Scheduling and Resource Allocation | 12 |
| Planning and Design | 10 |
| Control and Autonomous System Modeling | 10 |
| **Total** | 53 |

### B.3. Dataset Fields

Each problem entry is annotated with structured metadata to facilitate systematic evaluation. The fields and their descriptions are summarized in Table 4.

### B.4. Evaluation Dimensions

The evaluation framework is designed to capture the multifaceted requirements of engineering problem solving. It consists of four primary dimensions and one final criterion:

- **Information Extraction (IE):** Measures whether the model can correctly identify essential parameters such as values, objectives, and constraints.

- **Domain-Specific Reasoning (DR):** Evaluates the ability to apply specialized engineering knowledge to derive intermediate models or formulations.

*Table 4.* Description of dataset fields.

| Field | Description |
| --- | --- |
| ID | Unique identifier for each problem. |
| Problem | Natural-language description of the problem. |
| Reference | Citation of the original paper or publication from which the problem is derived. |
| Field | Category label, chosen from the four domains in Section A.1. |
| Information Extraction (IE) | Ability to identify numerical values, constraints, and objectives. |
| Domain-Specific Reasoning (DR) | Ability to apply engineering knowledge and logical inference. |
| Multi-Objective Decision-Making (MO) | Ability to balance multiple objectives such as cost, efficiency, and reliability. |
| Uncertainty Handling (UH) | Robustness against perturbations, incomplete information, or stochastic variations. |
| Feasibility | Indicator of whether the solution not only satisfies all explicit constraints but also remains consistent with the numerical data and conditions stated in the problem description. |

- **Multi-Objective Decision-Making (MO):** Assesses the capability of balancing multiple and potentially conflicting objectives.

- **Uncertainty Handling (UH):** Examines the robustness of solutions under incomplete or perturbed information.

- **Feasibility:** Serves as the ultimate criterion, determining whether solutions both satisfy all problem constraints and remain fully consistent with the given numerical data and conditions.

### B.5. Feasibility Evaluation Principles

In engineering tasks, the core purpose of *Feasibility* is to determine whether a proposed solution violates essential physical, logical, or operational constraints such that the solution becomes impossible to implement in practice. Feasibility evaluation does not require exhaustive descriptive completeness; rather, it examines whether there exists any violation that fundamentally invalidates the solution. A solution is therefore considered feasible only if all necessary constraints are strictly satisfied. Conversely, any violation of these constraints immediately leads to infeasibility.

The feasibility evaluation principles can be summarized as follows:

- **Evaluation Objective:** Check whether the proposed solution violates key physical conservation laws, safety limits, logical consistency, or implementable operational requirements.

- **Definition of Violations:** Any condition that breaks energy conservation, capacity limits, safety constraints, logical consistency, or mathematical solvability is regarded as infeasible.

- **Not Part of Feasibility Assessment:** Whether certain internal quantities are explicitly modeled (e.g., whether an intermediate variable is recorded), whether full derivation steps are shown, or whether all implementation details are specified does not affect feasibility as long as all constraints remain satisfied.

Thus, the essence of feasibility evaluation is to determine whether any critical constraint is violated, rather than requiring the solution to exhaustively specify every modeling detail.

**Example.** Consider the case of photovoltaic (PV) curtailment. One can explicitly model the unused PV energy through a variable $cur_{i,t}$ that appears in the energy balance constraint, or implicitly represent curtailment by allowing $g_{i,t}^{\mathrm{gen}}$ to operate

below its maximum bound, i.e.,

$$0 \leq g_{i,t}^{\text{gen}} \leq \overline{g}_{i,t},$$

without outputting a specific curtailment value. Both approaches are physically valid because PV units can always reduce their power output.

However, the implications for feasibility verification differ:

- **Explicit modeling:** If $cur_{i,t}$ is explicitly introduced, it must participate in energy conservation verification together with generation, demand, charging/discharging, and trade flows. Any inconsistency between the recorded curtailment and the actual energy balance would constitute a constraint violation and must be classified as infeasible.

- **Implicit modeling:** If no explicit curtailment variable is used, no additional verification is required; feasibility remains guaranteed as long as the generation upper bound is satisfied and the energy balance accounts for actual generation.

Hence, feasibility hinges on strict satisfaction of constraints, while explicit or implicit representation of internal quantities reflects only differences in modeling granularity, not differences in feasibility conditions.

### B.6. Annotation Standards and Expert Decision Protocol

To ensure the reliability, interpretability, and traceability of feasibility annotations in the EngiAgent dataset, we adopted a structured, fully human-driven labeling workflow involving more than 20 experienced domain experts. Each data sample was jointly evaluated by two annotators, who determined feasibility and extracted the essential hard constraints strictly based on explicit problem statements and associated numerical information. When disagreement or uncertainty arose, a third senior annotator with deeper domain experience (all holding a PhD degree) adjudicated the final conclusion to ensure consistency and reduce subjective interpretation bias. When necessary, annotators were allowed to reference fundamental physical laws, established engineering practices, and standard operational rules; however, personal modeling preferences or additional subjective assumptions were strictly prohibited to maintain objectivity and real-world applicability.

Importantly, feasibility labels were **not** generated automatically by LLMs. No LLM-based automated labeling mechanism was used. LLM tools were permitted only for auxiliary clarification tasks, such as terminology interpretation, and their outputs were **never** used as direct feasibility decisions or replacements for human judgment. All feasibility assessments were made solely by human experts based on the task description and explicit engineering constraints, independent of any model-generated solutions, ensuring a fully human-controlled and system-agnostic evaluation process.

To promote consistent reasoning and avoid divergence in interpretation, annotators were provided with a reference prompt serving solely as a cognitive guide rather than a decision rule or automated inference template.

**Inter-Annotator Disagreement and Resolution.** During the annotation process, a small number of instances (4 out of 53) initially received inconsistent feasibility judgments from the two primary annotators. These disagreements mainly stemmed from differing interpretations of implicit terminal conditions and constraint completeness (e.g., whether certain terminal state constraints such as terminal state-of-charge (SOC) should be explicitly enforced).

All disputed cases were subsequently reviewed through structured expert discussion involving the two annotators and a third senior domain expert. Through this adjudication process, a unified interpretation of the problem requirements was reached, and the essential feasibility constraints were clarified and consistently applied across the dataset. The finalized labels therefore reflect consensus conclusions grounded in explicit engineering principles and standardized constraint definitions.

The reference prompt is as follows:

---

**Reference Prompt for Feasibility Assessment**

Given the modeling problem, identify the essential constraints that must be satisfied to ensure feasibility. Focus only on the most critical mandatory conditions implied by the problem statement, taking into account both explicit limitations and general engineering knowledge or operational rules that must be respected. Present the constraints in a numbered list (1., 2., 3., ...), using clear and concise reasoning.

---

This protocol ensures that feasibility annotations are grounded in rigorous engineering reasoning and maintain transparency, consistency, and reproducibility for future research and dataset reuse.

## B.7. Example Adjudication and Error Analysis

**Problem Description.** In a simplified electricity market, two conventional generators, one wind farm, and one storage unit jointly supply demand over eight hourly periods. The conventional generators have block-based generation costs and ramping limits, while the wind farm provides zero-cost and time-varying output. The storage unit can charge and discharge subject to efficiency losses and limited energy capacity. Market clearing matches supply and demand, with prices determined by the marginal offer. The objective is to formulate a mathematical optimization model that determines hourly dispatch, storage scheduling, and resulting prices, and then analyze the effects of storage on (i) social welfare, (ii) wind curtailment, and (iii) firm profits.

| Unit | Block Cap. (MW) | Ramp Limit (MW/h) | Cost ($/MWh) | Notes |
|---|---|---|---|---|
| Conventional Gen 1 | 120, 120 | $+200 / -180$ | 30, 40 | Inc. marginal cost |
| Conventional Gen 2 | 80, 80 | $+150 / -130$ | 50, 60 | Inc. marginal cost |
| Wind Farm | $\leq 250$ (vary by hour) | N/A | 0 | Hourly avail. provided |
| Storage Unit | 200 MW (charge/discharge) | – | 0 | $SOC_0 = 100$, cap=200 MWh, $\eta = 0.85$ |

**System Data.** Hourly demand $D_t$ and wind availability $\overline{W}_t$ are explicitly provided for $t = 1, \ldots, 8$.

**Feasibility Constraint Set.** The following six constraints are annotated as **required hard feasibility conditions**:

1. Hourly power balance (for $t = 1, \ldots, 8$):

$$W_t + G_t^{(1)} + G_t^{(2)} + H_t = D_t + C_t.$$

2. Conventional generator block limits:

$$G_t^{(1)} = G_{t,1}^{(1)} + G_{t,2}^{(1)}, \quad 0 \leq G_{t,1}^{(1)} \leq 120, \ 0 \leq G_{t,2}^{(1)} \leq 120,$$
$$G_t^{(2)} = G_{t,1}^{(2)} + G_{t,2}^{(2)}, \quad 0 \leq G_{t,1}^{(2)} \leq 80, \ 0 \leq G_{t,2}^{(2)} \leq 80.$$

3. Conventional generator ramping limits (for $t = 2, \ldots, 8$):

$$G_t^{(1)} - G_{t-1}^{(1)} \leq 200, \qquad G_{t-1}^{(1)} - G_t^{(1)} \leq 180,$$
$$G_t^{(2)} - G_{t-1}^{(2)} \leq 150, \qquad G_{t-1}^{(2)} - G_t^{(2)} \leq 130.$$

4. Wind availability bound:

$$0 \leq W_t \leq \overline{W}_t.$$

5. Storage SOC dynamics, bounds, and initial condition:

$$SOC_t = SOC_{t-1} + \eta_c C_t - \frac{1}{\eta_d} H_t, \qquad 0 \leq SOC_t \leq 200,$$

$$SOC_0 = 100.$$

6. Storage power limits and charge/discharge exclusivity:

$$0 \leq C_t \leq 200 u_t, \quad 0 \leq H_t \leq 200 v_t, \quad u_t + v_t \leq 1, \quad u_t, v_t \in \{0, 1\}.$$

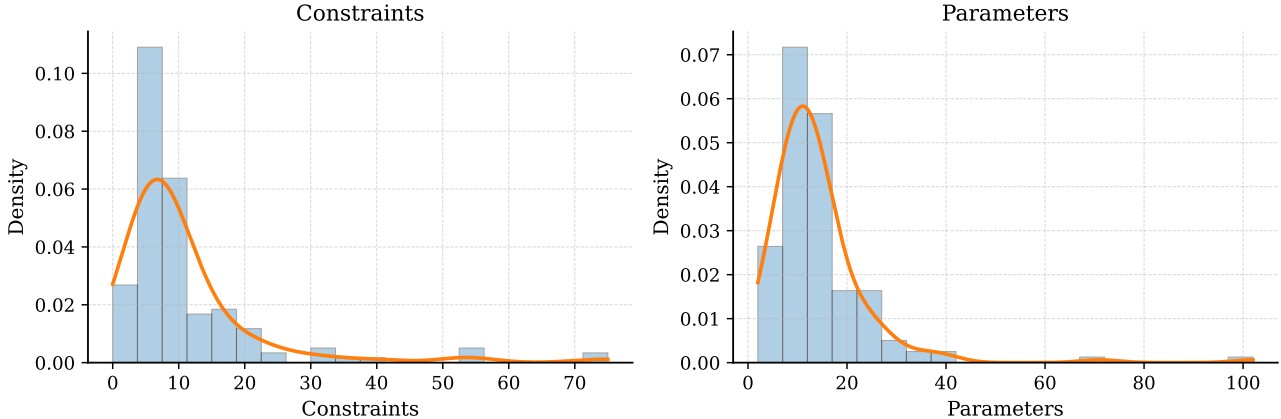

*Figure 7.* Distribution of task complexity in the EngiAgent benchmark. Left: density distribution of the number of constraints across all 53 tasks. Right: density distribution of the number of parameters. Both exhibit long-tailed behavior, indicating coverage from medium-scale to highly constrained engineering problems.

**Adjudication and Error Analysis.** All three annotators agreed that the above six constraints are mandatory feasibility conditions because they encode (i) energy conservation, (ii) device capability limits, and (iii) physically valid storage dynamics. During the annotation process, a disagreement occurred on whether a *terminal SOC condition* (e.g., $SOC_8 = SOC_0$ or $SOC_8 \geq SOC_0$) must also be included as a hard requirement. One annotator initially rated the problem as *infeasible* without such a terminal constraint, arguing that discharging to exhaustion may be undesirable in long-term planning. This created a potential **false-infeasible** labeling case.

After expert adjudication, it was clarified that terminal SOC requirements reflect *modeling policy or preference* rather than *physical feasibility necessity*, because the problem statement does not specify an operational-horizon sustainability requirement. Therefore, as long as Constraints (1)–(6) are satisfied, the problem admits a valid feasible solution space. The final feasibility label for P1 is recorded as **Feasible**. This example is documented as a representative case where over-restrictive assumptions could incorrectly induce a false-infeasible judgment.

### B.8. Complexity Distribution of Engineering Tasks

To further clarify the complexity of the engineering tasks used in EngiAgent, we report the empirical distribution of constraint and parameter scales over all tasks. As discussed in the main text, these tasks are open-ended: the benchmark specifies engineering objectives and physical or operational constraints, while the exact mathematical formulation (including the number of variables, the number of constraints, and solver class) is determined by how the LLM chooses to model each problem. Thus, complexity is not a predefined attribute of the task description, but an emergent property of the generated models.

Figure 7 shows the distributions of the number of constraints and the number of parameters across all generated solutions. Both metrics exhibit a clear long-tailed pattern: most tasks fall into a medium scale range (approximately 5–20 constraints), while a non-negligible subset reaches substantially higher complexity (e.g., more than 50 constraints or more than 100 parameters). This behavior is consistent with real-world engineering practice, where routine operational problems coexist with a smaller number of highly constrained, large-scale planning or coordination tasks. These results support our claim that the benchmark covers a broad spectrum of engineering difficulty, rather than a narrow set of simplified toy problems.

### B.9. Error-Type Diagnostics of Infeasible Solutions

To further understand the sources of infeasibility, we conduct a diagnostic analysis on DS-Agent with GPT-4o, where the gap between numerical solution rate and feasibility rate is the largest. Specifically, DS-Agent produces numerical solutions for 62.26% of the problems, but only 5.66% are feasible, indicating that many numerical outputs still violate essential engineering requirements.

We categorize infeasible solutions into four error types, following the taxonomy introduced in Section 3: (1) violating

physical laws, where solutions break fundamental engineering constraints such as capacity, conservation, or safety limits; (2) vague modeling, where the response describes high-level methods but fails to provide executable variables, equations, or constraints; (3) altering key data, where the solution changes critical numerical values or problem conditions; and (4) over-constraining the model, where unnecessary or unrealistic constraints eliminate feasible solutions.

The diagnostic results show that violating physical laws is the most frequent error type, accounting for 42.1% of infeasible cases. Vague modeling accounts for 36.8%, altering key data accounts for 15.8%, and over-constraining the model accounts for 5.3%. These results suggest that infeasibility is not merely caused by execution failure, but often arises from deeper violations of engineering semantics, data consistency, and physical validity.

## C. Core Prompt Design of EngiAgent

The effectiveness of EngiLLM relies heavily on carefully crafted prompts instructing each specialized agent to perform domain-specific tasks with precision and consistency. These prompts incorporate engineering domain knowledge, structured output formats, and error handling mechanisms to ensure reliable automated modeling and solving workflows. Each agent's prompt is designed to maintain coherence with the overall system architecture while addressing the unique requirements of their respective roles in the engineering problem-solving pipeline.

### C.1. Analyzer

The Analyzer prompt is designed to systematically extract and structure engineering problem information from natural language descriptions. It guides the LLM to identify decision variables, constraints, objectives, and implicit engineering rules while maintaining consistency with downstream modeling requirements. The prompt emphasizes comprehensive problem understanding and standardized JSON output generation to ensure seamless integration with the Modeler.

```
1  initial_modeling_analysis_prompt = """You are a senior engineering problem modeling expert
       . Please perform a **high-quality systematic analysis** of the following engineering
       problem, referring to the retrieved modeling knowledge to extract complete modeling
       elements.
2
3  **Problem Description:**
4  {problem_text}
5
6  **Retrieved Relevant Modeling Methods:**
7  - **Recommended Domain:** {hmml_analysis.get('domain', 'General Engineering')}
8  - **Subdomain:** {hmml_analysis.get('subdomain', 'Optimization')}
9  - **Recommended Method:** {hmml_analysis.get('method', 'Mathematical Programming')}
10 - **Confidence:** {hmml_analysis.get('confidence', 0.5):.2f}
11
12 **Modeling Guidance:**
13 - **Modeling Approach:** {modeling_guidance.get('approach', 'Mathematical optimization-
       based modeling')}
14 - **Core Concept:** {modeling_guidance.get('methodology', 'Construct objective function
       and constraints')}
15 - **Mathematical Framework:** {modeling_guidance.get('framework', 'Optimization framework
       ')}
16 - **Solution Strategy:** {modeling_guidance.get('solution_strategy', 'Numerical
       optimization methods')}
17
18 **Application Context:**
19 - **Typical Applications:** {hmml_result.get('application_context', {}).get('applications
       ', 'Engineering optimization problems')}
20 - **Method Advantages:** {hmml_result.get('application_context', {}).get('advantages', '
       Mature theory')}
21
22 ### Role and Mission ###
23 You are a top-tier "system architect" in charge of designing a complex, open-ended
       engineering problem into a **highly structured, traceable, and hierarchical JSON
       modeling blueprint**. This blueprint is the sole basis for downstream code generation
       agents to build accurate, feasible solving models.
24
25 ### Core Thinking Framework: Four Modeling Dimensions ###
```

```
26  When analyzing a problem, you must consider the following four dimensions as implicit
       indicators of your analysis. Your final output structure must reflect the results of
       this deep thinking process, but **do not** create top-level blocks in the JSON for
       these four dimensions.
27  1.  **Information Extraction (Information Extraction)**: Your goal is to identify all "
       atomic" information needed for modeling – entities, numerical values, relationships,
       and goals. This is the foundation of modeling.
28  2.  **Domain-specific Reasoning (Domain-specific Reasoning)**: You need to "dress" these "
       atomic" pieces of information with "domain clothing". Using professional engineering
       knowledge, transform raw data into meaningful parameters, convert relationships into
       physical or logical constraints, and select the most appropriate modeling paradigm.
29  3.  **Multi-objective Decision-making (Multi-objective Decision-making)**: You need to
       prioritize all goals. A **core optimization goal** must be clearly defined, while
       other goals are recognized as secondary or trade-off items. This is the "compass"
       guiding your decisions.
30  4.  **Uncertainty Handling (Uncertainty Handling)**: You need to identify all "cracks" in
       the information – missing, fuzzy, or variable parts. Your task is to "fill" these
       cracks through reasonable assumptions, ensuring the integrity and certainty of the
       core model.
31
32  ### Strict Output Format: Hierarchical JSON Modeling Blueprint ###
33  You must, and can only output a JSON object wrapped in ```json...```. This structure has
       been carefully designed to ensure the completeness, hierarchy, and executability of
       downstream agents.
34  ```json...
35  {
36    "modeling_context": {
37      "problem_essence": "Summarize the core engineering optimization problem in one
       sentence, formatted as 'Under [Key Constraints], optimize [Decision Variables] to
       achieve [Core Goals].".
38      "engineering_domain": "Specific engineering field (e.g., 'Power System Dispatch', '
       Supply Chain Network Design', 'Aerospace Planning')",
39      "modeling_paradigm": "Recommended core modeling methods based on problem
       characteristics and industry conventions (e.g., 'Mixed-Integer Linear Programming', '
       Quadratic Constraint Programming', 'Stochastic Optimization')",
40      "solution_scope": "Clearly define the core scope and boundary conditions of this
       modeling exercise."
41    },
42    "core_model_elements": {
43      "description": "Defines the minimum viable model of the problem (Minimum Viable Model)
       , including all necessary elements for constructing a basic solution method. This part
        must be self-consistent and complete, directly usable for code implementation.",
44      "decision_variables": [
45        {
46          "name": "Standard Mathematical Symbol Variable Name (e.g., P_g_t)",
47          "description": "Exact engineering meaning of the variable, index definition (e.g.,
       'g' represents generating units, 't' represents time periods) and unit",
48          "type": "continuous/integer/binary",
49          "domain": "Mathematical definition domain (e.g., '>= 0', '[0, 1]')",
50          "shape": "Dimension or size of the variable (e.g., '[G, T]' G is the number of
       units, T is the number of time periods)"
51        }
52      ],
53      "parameters": [
54        {
55          "param_id": "PARAM_01",
56          "name": "Mathematical Symbol of Parameter (e.g., C_g_max)",
57          "value": "Specific numerical values or complete arrays/lists from the original
       problem description. No omissions or descriptive substitutions are allowed.",
58          "unit": "Physical unit (e.g., 'MW', 'USD/MWh', 'kg')",
59          "description": "Exact engineering meaning and index explanation of the parameter
       .",
60          "source_reference": "Indicates the source or basis of this data in the original
       problem description (e.g., 'from Table 1, Column 3', 'Section 5 of the problem
       description explicitly states')"
```

```
61          }
62        ],
63       "objective_function": {
64         "name": "Name of the core objective function (e.g., TotalOperatingCost)",
65         "type": "minimize/maximize",
66         "expression": "Complete mathematical expression using the defined variable and
      parameter symbols.",
67         "components": [
68           {
69             "component_expr": "One part of the expression (e.g., sum(c_g * P_g_t for g, t))
      ",
70             "description": "Engineering meaning of this component (e.g., 'Total Fuel Cost')"
71           }
72         ]
73       },
74       "constraints": [
75         {
76           "constraint_id": "CONST_01",
77           "name": "Unique Identifier of Constraint (e.g., 'PowerBalance')",
78           "expression": "Complete mathematical (equation/inequality) expression using the
      defined variable and parameter symbols.",
79           "category": "Category of the constraint ('Physical Laws', 'Resource Capacity', '
      Supply-Demand Balance', 'Operational Logic', 'Strategic Requirements')",
80           "description": "Engineering significance and importance of the constraint."
81         }
82       ]
83     },
84     "extended_analysis_and_robustness": {
85       "description": "Includes supplementary, expanded, and considerations under uncertainty
       for the core model, which is key to advanced analysis and ensuring robustness of
      solutions.",
86       "key_assumptions": [
87         {
88           "assumption_id": "ASSUM_01",
89           "content": "Explicit assumptions made to handle missing information or simplify
      the model.",
90           "justification": "Engineering principles, industry conventions, or logical basis
      for this assumption.",
91           "impact_on_model": "Explicitly states how this assumption specifically affects a
      specific element in `core_model_elements` (e.g., 'Setting the value of PARAM_05 `
      line_efficiency` to 0.98', 'Simplifying the calculation formula of CONST_03 `PowerFlow
      `')"
92         }
93       ],
94       "uncertainty_sources": [
95         {
96           "source_id": "UNCERT_01",
97           "description": "Key sources of uncertainty identified (e.g., 'Future Market
      Electricity Price', 'Equipment Failure Rate')",
98           "affected_elements": ["IDs of parameters or variables affected by this uncertainty
       (e.g., 'PARAM_10')"],
99           "handling_strategy": "Recommended strategies for handling (e.g., 'Using Expected
      Values for Deterministic Modeling', 'Conducting Sensitivity Analysis', 'Using Scenario
       Analysis Approach')
100        }
101      ],
102      "trade_off_analysis": {
103        "secondary_objectives": [
104          {
105            "name": "Secondary or potential optimization goals (e.g., '
      MinimizeCarbonEmissions')",
106            "expression": "Mathematical expression.",
107            "conflict_with": "Which core or secondary goals are in tension with each other (
      e.g., 'TotalOperatingCost')"
108          }
```

```
109          ],
110        "soft_constraints": [
111          {
112            "name": "Soft constraints or preferences (e.g., 'PreferredMaintenanceWindow')",
113            "description": "Conditions hoped for but not necessarily required, often modeled
       by adding penalty terms to the objective function."
114          }
115        ]
116      },
117      "sensitivity_factors": [
118        {
119          "param_id": "ID of key parameters for conducting sensitivity analysis (from '
       parameters' section)",
120          "justification": "Why this parameter might have a significant impact on the model
       results."
121        }
122      ]
123    }
124 }
125 ```
126
127 ### Golden Command and Final Verification ###
128 1.  **Absolute Data Integrity**: Every valid numerical value in the original problem **
       must** be present in the 'parameters' list with a unique entry, and its origin must be
        traceable through the 'source_reference' field.
129 2.  **Mandatory Internal References**: Cross-references within JSON are mandatory. For
       example, 'sensitivity_factors' must reference 'param_id' of 'parameters'; '
       key_assumptions''s 'impact_on_model' must explicitly point to a specific element in '
       core_model_elements'. This ensures traceability and consistency of the model.
130 3.  **Core Model Priority**: 'core_model_elements' is the cornerstone. It must be a model
       that can be independently solved and clearly defined. All supplementary, trade-off,
       and uncertainty analyses are based on it in 'extended_analysis_and_robustness'.
131 4.  **Clear Roles and Responsibilities**: Strictly differentiate between '
       core_model_elements' (what they do) and 'extended_analysis_and_robustness' (how to do
       it better/more robustly). Do not mix in assumptions or uncertain elements in the core
       model.
132 5.  **Designed for Code Generation**: All mathematical expressions ('expression') must use
        standard, unambiguous mathematical notation so that downstream agents can easily
       parse and convert them into code. """
```

### C.2. Modeler

The Modeler prompt focuses on translating structured problem formulations into executable Pyomo optimization code. It employs domain-specific modeling patterns, syntax validation guidelines, and common pitfall avoidance strategies tailored for engineering optimization open questions. The prompt ensures mathematical rigor while maintaining code clarity and computational efficiency for various problem types including linear, mixed-integer, and nonlinear programming.

```
1 basic_model_code_generation_prompt = f"""### Role & Mission ###
2 You are a top-tier code generation engine specialized in translating structured
     engineering modeling blueprints (JSON) into executable, industrial-grade Python code.
3
4 ### Core Instructions ###
5 Your **ONLY** task is to generate complete, accurate, directly executable Python solving
     code based on the 'MODELING_BLUEPRINT' JSON provided below. Every key-value pair in
     the JSON is a mandatory instruction that must be strictly followed.
6
7 ### Input: Modeling Blueprint (MODELING_BLUEPRINT) ###
8 ```json
9 {analysis_result_str}
10 ```
11
12 ### CRITICAL Data Structure & Dimension Rules ###
13 **NEVER use placeholder data like [...] or ellipsis**
14 **Always provide complete, specific data arrays**
```

```
15  - If JSON contains "[...]" or "...", replace with realistic example data
16  - Match array dimensions exactly to variable dimensions
17  - For time series: use 1D arrays like [0.1, 0.15, 0.2, ...]
18  - For scenarios: create 2D arrays like [[value1, value2], [value3, value4], ...]
19
20  **NEVER mismatch array dimensions with variable indices**
21  **Ensure parameter arrays match variable indexing**
22  ```python
23  # Wrong: Using 1D array with 3D variables
24  Electricity_price_t = [0.1, 0.15, ...]  # 1D
25  model.P_ev_t_s[e, t, s]  # 3D - MISMATCH!
26
27  # Correct: Match dimensions or simplify variables
28  # Option 1: Simplify to 2D variables
29  model.P_ev_t[e, t]  # 2D
30  Electricity_price_t = [0.1, 0.15, ...]  # 1D - MATCH!
31
32  # Option 2: Create appropriate multi-dimensional data
33  Electricity_price_t_s = [[0.1, 0.12], [0.15, 0.17], ...]  # 2D
34  model.P_ev_t_s[e, t, s]  # 3D with proper indexing
35  ```
36
37  ### CRITICAL Pyomo Summation Rules ###
38  **NEVER use summation() from pyomo.environ - it's error-prone**
39  **Always use Python's built-in sum() with generator expressions**
40  ```python
41  # Wrong: Using summation() function
42  from pyomo.environ import summation
43  electricity_cost = summation(Electricity_price_t, model.P_ev_t)  # ERROR!
44
45  # Correct: Using sum() with proper indexing
46  electricity_cost = sum(Electricity_price_t[t-1] * model.P_ev_t[e, t]
47                         for e in E for t in T)
48  ```
49
50  **NEVER pass arrays directly to sum() - always use explicit indexing**
51  **Always iterate over indices and access array elements**
52  ```python
53  # Wrong: Direct array operations in sum
54  sum(price_array * model.var for ...)  # ERROR!
55
56  # Correct: Index-based access
57  sum(price_array[i] * model.var[i] for i in range(len(price_array)))
58  ```
59
60  ### CRITICAL Pyomo Syntax Rules ###
61  **NEVER use Python built-ins in constraints**: max(), min(), abs()
62  **NEVER use if statements with Pyomo variables**: if model.x[i] >= 0: ...
63  **NEVER compare Pyomo variables in boolean context**: if P_ev_t[ev, t] >= 0
64  **NEVER use ternary operators with Pyomo variables**: value = model.x[i] if model.x[i] >=
        0 else 0
65  **NEVER mix Pyomo expressions with conditional logic**: model.x[i] * (1 if condition else
        0)
66  **For conditional logic**: Use separate variables or Pyomo Piecewise functions
67  **For max constraints**: Use auxiliary binary variables with Big-M method
68  **For absolute value**: Use two inequality constraints: x >= y and x >= -y
69
70  ### MOST COMMON ERRORS TO AVOID ###
71  **"Cannot convert non-constant Pyomo expression to bool" - CAUSED BY:**
72  ```python
73  # WRONG - These will cause fatal errors:
74  if model.P_s_t[t] >= 0:
75      return model.P_s_t[t] /
76  value = model.x[i] if model.x[i] > 0 else model.x[i] * 0.9
77  model.constraint = Constraint(expr=model.var[i] >= (5 if condition else 3))
```

```
78  ```
79
80  **CORRECT alternatives:**
81  ```python
82  # Method 1: Separate variables
83  model.P_charge = Var(T, domain=NonNegativeReals)
84  model.P_discharge = Var(T, domain=NonNegativeReals)
85
86  # Method 2: Binary variables with constraints
87  model.mode = Var(T, domain=Binary)
88  model.cons1 = Constraint(T, rule=lambda m,t: m.P[t] <= M * m.mode[t])
89  model.cons2 = Constraint(T, rule=lambda m,t: m.P[t] >= -M * (1 - m.mode[t]))
90  ```
91
92  ### CRITICAL: NEVER GENERATE INCOMPLETE CODE ###
93  **ABSOLUTELY FORBIDDEN**: Any code containing only "results = []" or similar empty
        placeholders
94  **ABSOLUTELY FORBIDDEN**: Comments like "# The rest of the code remains unchanged"
95  **ABSOLUTELY FORBIDDEN**: Any response that does not contain a COMPLETE Pyomo model
96
97  **MANDATORY REQUIREMENT**: Every response MUST contain a FULLY FUNCTIONAL Pyomo
        optimization model that can be executed immediately.
98
99  **Your code MUST include ALL of these components (NO EXCEPTIONS):**
100 1. Complete imports: `from pyomo.environ import *`
101 2. All parameter data with realistic values (no [...] placeholders)
102 3. Complete ConcreteModel() definition with ALL variables from JSON
103 4. Complete Objective function (maximize/minimize something meaningful)
104 5. ALL constraint definitions from the JSON properly implemented
105 6. Complete solver setup and solve() call
106 7. Result extraction and printing
107
108 **VERIFICATION CHECKLIST - Your code must pass ALL these checks:**
109 Does it import pyomo.environ?
110 Does it create a ConcreteModel()?
111 Does it define ALL variables mentioned in the JSON?
112 Does it implement ALL constraints from the JSON?
113 Does it define a meaningful objective function?
114 Does it call a solver and get results?
115 Is it longer than 100 lines of actual code?
116
117 ### Output Requirements ###
118 - Format: Output only one complete Python code block wrapped in ```python...```
119 - Content: Code must include all necessary library imports, parameter definitions, model
        construction, solver calls, and clear result output
120 - Quality: Code must be robust with basic solver status checks
121 - No extra explanation: Except for the code itself and necessary comments, do not add any
        preamble or summary
122 - Solver setup: Include intelligent solver selection (glpk, cbc, ipopt) with timeouts and
        error handling
123 - Data integrity: All parameter arrays must be complete with realistic values, no
        placeholders
124 """
```

### C.3. Verifier

The Verifier prompt establishes comprehensive quality assurance protocols for semantic verification between problem descriptions and generated models. It guides the LLM to perform multi-dimensional consistency checks, identify logical inconsistencies, and provide detailed feedback for model refinement. The prompt emphasizes distinguishing between critical errors and acceptable simplifications while maintaining alignment with original engineering problem requirements. Notably, the prompt integrates a dynamic tolerance adjustment mechanism that automatically relaxes verification standards when consecutive verification failures are detected, promoting system convergence while preventing inefficient excessive retry loops.

```
1  # Dynamically adjust Verification tolerance
2              tolerance_adjustment = ""
3              if self.consecutive_verification_failures > 8:
4                  tolerance_adjustment = """
5  ** Special Notice: Multiple verification loops detected, please adopt more lenient
        verification standards**
6  - For complex game theory/bilevel optimization problems, allow reasonable modeling
        simplification
7  - For MPEC/KKT conditions, accept pragmatic-oriented approximate implementations
8  - Focus on verifying core logic correctness, be tolerant of technical details
9  - Only judge as mismatch when fundamental errors exist"""
10             elif self.consecutive_verification_failures > 5:
11                 tolerance_adjustment = """
12 **Notice: Verification difficulties detected, please appropriately relax verification
        standards**
13 - Maintain understanding and tolerance for modeling strategy differences
14 - Focus on substantial issues, ignore technical implementation details"""
15
16             prompt = f"""### Role and Mission ###
17 You are a top-tier, **extremely pragmatic** engineering system verification expert. Your
        core mission is to serve as the system's "gatekeeper". Your task is not to conduct
        line-by-line code auditing, but to judge from a **system modeling perspective**
        whether the provided Python code faithfully solves the original engineering problem in
         terms of **core logic and key objectives**. You must distinguish between **modeling
        strategy differences** and **substantial logical errors**.
18 {tolerance_adjustment}
19
20 Your primary principle is **"Promote Convergence"**:
21 - **Default Trust**: Unless the code contains **catastrophic, directional** errors, it
        should default to `PASS`.
22 - **Understand Compromise**: Must recognize that the current code may be a **necessary
        compromise or simplification** made to solve **previous failures (such as overly
        strict constraints, solver infeasibility, etc.)**.
23 - **Minimal Intervention**: Your role is to confirm the overall direction is correct and
        the core has not deviated, not to micromanage technical details.
24
25 ### Input ###
26 1.  **[Problem Specification] Original Engineering Problem (The Problem Specification)**:
27     ```
28     {safe_original_problem}
29     ```
30 2.  **[Solution Implementation] Engineering Model Code (The Implemented Solution)**:
31     ```python
32     {safe_pyomo_code}
33     ```
34
35 ### Core Verification Instructions: Judgment Hierarchy Based on Historical Context ###
36 Please strictly follow this judgment logic to ensure your verification is context-aware.
37
38 **Step 1: Check for "Catastrophic" Errors (Deal-Breaker Check)**
39 This is the only reason you can judge as `FAIL`. Check if any of the following situations
        exist:
40 1.  **Objective Direction Reversed**: Minimization problems implemented as maximization,
        or vice versa.
41 2.  **Core Physical/Economic Law Violations**: Complete omission of constraints that
        define the problem foundation, such as "supply-demand balance", "energy conservation",
         etc.
42 3.  **Key Decision-Making Entities Missing**: In an obvious multi-agent game problem, the
        code completely fails to reflect interactions or decisions between different entities
        (even in simplified form).
43
44 *   If any of the above is found, immediately judge as `FAIL` and stop subsequent checks.
45 *   If none are found, **proceed directly to Step 2 and finally judge as `PASS`**.
46
```

```
47  **Step 2: Identify and Acknowledge "Pragmatic Deviations" (Pragmatic Deviation Recognition
       )**
48  When code passes Step 1 checks, it will be judged as 'PASS'. Your task is to identify
       differences between the current code and the original problem specification, and judge
        whether this is a reasonable response to errors in the '[Historical Context]'.
49
50  Common **reasonable deviations that should be accepted as 'PASS'** include:
51  -   **Relaxing/removing constraints to solve infeasible problems**: Should consider that
       previous reasonable modeling might have been "infeasible", and the current code
       relaxing or removing certain strict constraints is a **positive, should-be-encouraged
       ** moderate compromise.
52  -   **Simplifying models to address LLM capability limitations or solving complexity**:
       Simplifying complex bilevel optimization, MPEC or equilibrium models to appropriate
       single-level optimization, as long as key economic incentives or trade-off
       relationships are embodied through parameters or proxy constraints, should be accepted
       .
53  -   **Technical adjustments**: Adjusting variable boundaries, parameter units, Big-M
       coefficients, solver options, etc., to improve numerical stability or find feasible
       solutions.
54  -   **Minor discrepancies in data or indices**: Should be ignored as long as they don't
       affect the core model logic and magnitude.
55
56  ### Output Format ###
57  Please strictly return your judgment in the following JSON format, wrapped with '''json
       ...'''. **When the model adopts "acceptable approximation", please provide minimal
       incremental correction suggestions in 'suggestion' rather than outright rejection.**
58
59  '''json
60  {{
61    "mismatch_detected": true/false,
62    "mismatch_reason": "Specific mismatch reason",
63    "suggestion": "Fix or optimization suggestions",
64    "confidence": 0.0-1.0
65  }}
66  '''
67  """
```

# D. Case Study

To further illustrate the generality of EngiAgent across different engineering domains, we next present a second, complete example from the renewable energy domain, covering problem formulation, modeling, and solution generation.

---

**Example: Rooftop PV Layout Optimization (P35)**

**Problem Description**

We consider a rooftop photovoltaic (PV) layout optimization problem. The objective is to minimize the total capital expenditure (Capex) while ensuring that the annual electricity demand $D_{\text{tar}} = 50{,}000$ kWh/yr is satisfied. The system consists of a single flat roof with usable area $A_{\text{use}} = f_{\text{use}} \cdot A_{\text{roof}} = 420$ m$^2$. All panels must share one common tilt–azimuth configuration, chosen from a small set of candidates.

**Data**

The parameters of the rooftop system, available PV panel types, and candidate configurations are summarized in Tables 5 and 6.

---

*Table 5.* PV panel specifications.

| Panel Type | $P_i$ (kWp) | $a_i$ (m$^2$) | $c_i$ (€/panel) |
|---|---|---|---|
| Canadian 275 W | 0.275 | 1.64 | 130 |
| Jinko 330 W | 0.330 | 1.94 | 155 |
| Canadian 330 W | 0.330 | 1.94 | 175 |

*Table 6.* Candidate tilt–azimuth configurations and yields.

| Config $k$ | Tilt (°) | Azimuth (°) | $y_k$ (kWh/kWp·yr) |
|---|---|---|---|
| 1 | 20 | 160 | 1350 |
| 2 | 30 | 180 | 1450 |
| 3 | 35 | 135 | 1400 |

**Model Formulation**

**Decision Variables.**

- $n_i \in \mathbb{Z}_{\geq 0}$: number of panels of type $i$,

- $z_k \in \{0, 1\}$: binary variable for configuration choice,

- $K \geq 0$: installed capacity (kWp),

- $E_k \geq 0$: annual energy (kWh) for configuration $k$.

**Objective.**

$$\min \quad \text{Capex} = \sum_{i=1}^{3} c_i n_i + \varepsilon \sum_{i=1}^{3} n_i,$$

where $\varepsilon \ll \min_i c_i$ is a tie-breaker favoring fewer panels.

**Constraints.**

$$\sum_{k=1}^{3} z_k = 1, \qquad\qquad z_k \in \{0, 1\} \quad \text{(Single configuration)} \tag{1}$$

$$K = \sum_{i=1}^{3} P_i n_i, \qquad\qquad n_i \in \mathbb{Z}_{\geq 0} \quad \text{(Installed capacity)} \tag{2}$$

$$E_k \geq D_{\text{tar}} \cdot z_k, \qquad\qquad \forall k \quad \text{(Demand satisfaction)} \tag{3}$$

$$\sum_{i=1}^{3} a_i n_i \leq A_{\text{use}}, \qquad\qquad \text{(Usable roof area)} \tag{4}$$

$$E_k \leq y_k K + M(1 - z_k), \qquad \forall k \quad \text{(Linearized yield coupling)} \tag{5}$$

Optional constraints (not used in this case) include:

$$\sum_{i=1}^{3} a_i n_i \geq \alpha_{\min} A_{\text{use}}, \quad \alpha_{\min} \in [0, 1] \quad \text{(Coverage lower bound)} \tag{6}$$

$$n_i \leq \overline{n}_i, \quad i = 1, 2, 3 \quad \text{(Panel availability limit)} \tag{7}$$

**Optimal Solution**

The MILP was solved using GLPK. The optimal feasible solution is:

- Selected configuration: $k = 2$ (Tilt 30°, Azimuth 180°),

- Panel counts: $n_1 = 3$, $n_2 = 102$, $n_3 = 0$,

- Installed capacity: $K = 34.49$ kWp,

- Annual energy yield: $E = 50,000$ kWh,

- Capital expenditure: €16,200.

**Performance Metrics**

$$S = p_{\text{el}} \cdot E = 9,000/\text{yr}, \qquad \text{Payback} = \frac{\text{Capex}}{S} \approx 1.8 \text{ years},$$

$$\text{CO}_{2,\text{avoided}} = \gamma_{\text{CO2}} \cdot E = 12,500 \text{ kg/yr}, \qquad \text{Roof coverage} = \frac{\sum_i a_i n_i}{A_{\text{use}}} \approx 49.6\%.$$

**Pyomo Implementation**

```
from pyomo.environ import *

# Initialize the Pyomo model
model = ConcreteModel()

# Define sets
model.I = RangeSet(3)   # panel types
model.K = RangeSet(3)   # configurations

# Parameters
params = {
    'A_roof': 600,
    'f_use': 0.7,
    'A_use': 420,
    'p_el': 0.18,
    ' _CO2 ': 0.25,
    'D_tar': 50000,
    'P_i': [0.275, 0.33, 0.33],
    'a_i': [1.64, 1.94, 1.94],
    'c_i': [130, 155, 175],
    'y_k': [1350, 1450, 1400],
    '  ': 1e-05,
    'M': 100000
}

# Variables
model.n_i = Var(model.I, domain=NonNegativeIntegers)
model.z_k = Var(model.K, domain=Binary)
model.K_installed = Var(domain=NonNegativeReals)
model.E_k = Var(model.K, domain=NonNegativeReals)

# Objective
model.MinimizeCapex = Objective(
    expr=sum(params['c_i'][i-1]*model.n_i[i] for i in model.I) +
         params['  ']*sum(model.n_i[i] for i in model.I),
    sense=minimize
)

# Constraints
def energy_demand_rule(m, k):
```

```
41       return m.E_k[k] >= params['D_tar'] * m.z_k[k]
42 model.EnergyDemand = Constraint(model.K, rule=energy_demand_rule)
43
44 model.RoofArea = Constraint(
45     expr=sum(params['a_i'][i-1]*model.n_i[i] for i in model.I) <= params['A_use']
46 )
47
48 model.ConfigurationChoice = Constraint(
49     expr=sum(model.z_k[k] for k in model.K) == 1
50 )
51
52 model.InstalledCapacity = Constraint(
53     expr=model.K_installed == sum(params['P_i'][i-1]*model.n_i[i] for i in model.I)
54 )
55
56 def linearize_energy_yield_rule(m, k):
57     return m.E_k[k] <= params['y_k'][k-1]*m.K_installed + params['M']*(1-m.z_k[k])
58 model.LinearizeAnnualEnergyYield = Constraint(model.K, rule=
     linearize_energy_yield_rule)
59
60 # Solver
61 solver = SolverFactory('glpk')
62 results = solver.solve(model, tee=True)
63
64 print("Optimal Capex:", model.MinimizeCapex())
65 print("Panel counts:", [model.n_i[i].value for i in model.I])
66 print("Selected config:", [model.z_k[k].value for k in model.K])
67 print("Installed capacity (kWp):", model.K_installed())
68 print("Annual energy (kWh):", [model.E_k[k].value for k in model.K])
```

## E. Coordinator Routing Trace Examples

This appendix provides five representative coordinator routing traces to demonstrate that the proposed Fully Connected Coordinator performs interpretable, auditable, and debugging-oriented agent assignment rather than opaque heuristic switching. Each routing action records: (i) the detected trigger signal, (ii) the natural-language reasoning trace, (iii) the actionable strategy, (iv) the selected agent, (v) the estimated internal success likelihood, and (vi) the execution outcome. These traces exemplify how the system performs iterative diagnosis, refinement, and verification toward feasible engineering solutions.

---

**Routing Trace Example 1 (Real Extract)**

**Trigger:** Import failure during Pyomo model construction.

**Routing reasoning:**
```
trigger_signal:  ImportError:  missing attribute 'suffixes' from 'pyomo.environ'
target_agent:  modeler
strategy:  Remove invalid import and update Pyomo import statements
priority:  high
estimated_success_rate:  0.40
should_continue:  true
reasoning:  The failure originates from an incorrect import.  This is a structural
modeling issue rather than conceptual or optimization-phase related; the Modeler
agent is responsible for code-level construction.
```

**Debug log:**
```
Record debugging attempt:  modeler - ImportError - failure
Record debugging attempt:  modeler - success - success
```

---

---

### Routing Trace Example 2 (Structural Feasibility Refinement)

**Trigger:** Detected non-linear structure after initial formulation.

**Routing reasoning:**
```
trigger_signal:  Model violates linear-programming structure due to non-linear SoC
constraint
target_agent:  analyzer
strategy:  Re-inspect SoC constraint formulation and propose an LP-compatible
alternative
priority:  medium
estimated_success_rate:  0.55
should_continue:  true
reasoning:  The solver reports nonlinearity arising from a division term in SoC
dynamics.  This is a formulation correctness issue that requires a domain-level
reasoning pass before code regeneration.
```

**Debug log:**
```
Record debugging attempt:  analyzer – StructuralNonlinearity – success
```

---

### Routing Trace Example 3 (Semantic Interpretation Correction)

**Trigger:** Solver returned infeasible status for bus-level nodal balance.

**Routing reasoning:**
```
trigger_signal:  Infeasibility:  Bus-2 nodal balance violated across scenarios
target_agent:  verifier
strategy:  Locate imbalance sources, inspect demand-to-bus mapping semantics
priority:  medium
estimated_success_rate:  0.63
should_continue:  true
reasoning:  The variables d_j represent served demand but were not properly mapped
to the bus topology.  This indicates semantic--not syntactic--misalignment.
```

**Debug log:**
```
Record debugging attempt:  verifier – InfeasibilityMismatch – failure
```

---

### Routing Trace Example 4 (Directed Model Refinement)

**Trigger:** Semantic inconsistency confirmed, directed fix needed.

**Routing reasoning:**
```
trigger_signal:  Verifier confirmed mismatch in demand-to-bus linkage
target_agent:  modeler
strategy:  Rewrite nodal balance equations using explicit bus-indexed demand sets
priority:  high
estimated_success_rate:  0.72
should_continue:  true
reasoning:  The root cause is localizable and fixable at the model construction
```

```
layer; no conceptual redesign is required.

Debug log:
Record debugging attempt: modeler - ConstraintReformulation - success
```

**Routing Trace Example 5 (Final Validation and Commitment)**

**Trigger:** Feasible solution obtained; final validation pending.

**Routing reasoning:**
```
trigger_signal: Feasible solution obtained but pending specification-level
validation
target_agent: solution_verifier
strategy: Validate scenario robustness, SoC boundary satisfaction, and objective
semantics
priority: high
estimated_success_rate: 0.89
should_continue: false
reasoning: All constraints are satisfied; a final conformance check is required
before commitment.

Debug log:
Record debugging attempt: solution_verifier - FinalValidation - success
```

These five examples collectively demonstrate that the proposed Fully Connected Coordinator supports multi-step diagnosis and refinement, records interpretable decision rationales, and provides actionable, traceable debugging history rather than opaque routing behavior.

## F. Baseline Fairness Summary

To ensure transparent and fair comparison across agent frameworks, we summarize the adaptation policies used for all baselines in the EngiAgent evaluation:

- **Unified solver access without enhancing baseline capabilities.** All baselines are granted access to the same numerical solver used by EngiAgent. We do not provide additional templates, verification logic, or domain-specific optimizations. Thus, performance differences arise from reasoning and modeling quality, not from tooling advantages.

- **Minimal enabling fixes only.** When a baseline fails due to syntax errors, unbound variables, or missing numeric values, we apply only the minimum fixes necessary to make it executable. These corrections do not alter its modeling logic or constraints and therefore do not improve its solution quality.

- **Original domain prompts are preserved.** Domain hints are part of each method's intended design. Forcing a shared prompt would alter behavior and artificially weaken baselines. We therefore preserve original prompting strategies and only unify solver access.

- **Transparent compute budget reporting instead of forced equality.** Since agent architectures differ substantially, forcing unified budgets would unfairly penalize some methods. We therefore run all baselines using their recommended configurations and report runtime, token usage, and cost in Sec. 6.4 to enable a transparent cost–performance audit.

## G. Framework Extensibility and Cross-Tool Capability

This appendix presents two additional benchmark tasks (P7 and P40) to illustrate that EngiAgent is not restricted to Pyomo-based modeling nor dependent on optimization pipelines or solver-specific behavior. While Pyomo is used in part of

the benchmark as a unified feasibility and solver interface, it is not required for model construction; the feasibility-driven coordination mechanism remains unchanged. As shown in Section G.1, EngiAgent operates in the same way on ML–based tasks that treat optimization only as a supporting tool (P7), and as demonstrated in Section G.2, it also handles tasks that involve no optimization or modeling stack at all (P40). These results serve as initial demonstrations that only lightweight adjustments to the Modeler/Analyzer prompts and minor changes to solver configuration and result extraction are sufficient to support heterogeneous modeling paradigms, without modifying the core architecture.

### G.1. ML Regression with MILP Feature Search

This task builds a regression forecasting model evaluated using Mean Absolute Percentage Error (MAPE). The regression form is constructed through machine-learning numerical estimation, and a mixed-integer linear program (MILP) is used solely for feature selection. Thus, optimization acts only as a supporting mechanism for exploring model configurations, rather than as the representation of domain logic. Pyomo is used only as a generic solver wrapper; the same formulation can alternatively be implemented via OR-Tools, JuMP, or direct solver APIs without affecting agent behavior or feasibility coordination.

**Core implementation excerpt (ML + MILP; solver-agnostic):**

```
1  # ML regression form (constructed by ML semantics, not symbolic modeling)
2  model.beta_0 = Var(domain=Reals)
3  model.beta_j = Var(range(k), domain=Reals)
4  model.delta_j = Var(range(k), domain=Binary)  # Feature selection decision
5
6  # Aux term for product beta_j * delta_j (linearized MILP, tool-agnostic)
7  model.beta_j_delta = Var(range(k), range(n), domain=Reals)
8  for j,t in product(range(k), range(n)):
9      model.beta_j_delta[j,t] <=  M * model.delta_j[j]
10     model.beta_j_delta[j,t] >= -M * model.delta_j[j]
11     model.beta_j_delta[j,t] >=  model.beta_j[j] - M*(1-model.delta_j[j])
12     model.beta_j_delta[j,t] <=  model.beta_j[j] + M*(1-model.delta_j[j])
13
14 # Regression prediction used directly by Analyzer & Verifier
15 def pred_rule(m, t):
16     return m.y_hat[t] == m.beta_0 + sum(
17         m.beta_j_delta[j,t] * X[j,t] for j in range(k)
18     )
19 model.pred = Constraint(range(n), rule=pred_rule)
20
21 # MAPE objective (supports ML evaluation, not domain construction)
22 model.obj = Objective(
23     expr=(100/n)*sum(model.abs_error[t]/Y[t] for t in range(n))
24         + lambda_cost*sum(model.delta_j[j] for j in range(k))
25 )
26
27 # Any MILP solver may be used; Pyomo only wraps the call
28 results = SolverFactory("cbc").solve(model)
```

### G.2. Pure ML Forecasting without Optimization Solvers

This task performs forecasting using numerical machine-learning routines, without constructing an explicit optimization model via modeling languages, invoking a solver, or relying on modeling frameworks. Feasibility is established entirely through statistical and data-consistency checks conducted by EngiAgent's Verifier and handled through the same error-routing mechanism.

**Core implementation excerpt (no solver, no modeling stack):**

```
1  # Closed-form OLS estimation (pure numerical ML; no solver backend)
2  X = np.column_stack([np.ones(n), X1, X2, X3])
3  theta = np.linalg.inv(X.T @ X) @ X.T @ Y
4
5  # Prediction computed entirely by ML numerical routines
6  y_hat = X @ theta
```

```
7
8  # Feasibility validation (handled by Verifier, independent of solvers)
9  if np.any(np.isnan(theta)) or np.any(np.isinf(theta)):
10     raise ValueError("Infeasible ML estimation")
```

### G.3. Summary and Future Directions

These extended cases show that EngiAgent supports both solver-dependent ML optimization modeling (e.g., feature search) and solver-free ML tasks (e.g., closed-form OLS estimation) without depending on Pyomo, symbolic modeling frameworks, or specific solver backends. The core architecture remains unchanged; only tool-level interactions require minor adjustments in prompts and output handling.

This tool-agnostic coordination suggests a natural pathway toward broader engineering domains, such as PDE-based simulation, stochastic system modeling, and hybrid data–physics applications, where specialized estimators or solvers can be introduced while maintaining the same feasibility-centered agent workflow.

