# OpenReview forum: "EngiAgent: Fully Connected Coordination of LLM Agents for Solving Open-ended Engineering Problems with Feasible Solutions"
_ICML.cc/2026/Conference — ICML 2026 regular_

### Official Review · Reviewer_Z7AS · 2026-02-19

**Soundness:** 3
**Presentation:** 3
**Significance:** 2
**Originality:** 3
**Overall Recommendation:** 4
**Confidence:** 3

**Summary:**

The paper introduces EngiAgent, an LLM-driven multi-agent framework designed to overcome the weakness of LLMs to generate mathematically plausible but physically unfeasible engineering solutions. This paper propose to utilize a fully connected coordinator that dynamically routes feedback between specialized agents focused on analysis, modeling, and verification. This architecture mimics professional expert workflows, allowing the system to iteratively refine models and self-correct for data inconsistencies or solver failures. By testing across four diverse domains, the authors demonstrate that this flexible, feasibility-oriented approach significantly outperforms other LLM-based methods, improving LLM-driven multi-agents a more reliable tool for complex, real-world engineering constraints.

**Compliance With Llm Reviewing Policy:**

Affirmed.

**Final Justification:**

I originally thought this work is primarily a prompt engineering work, with very large claim of solving open engineering problem.

After rebuttal, I understand the importance of fully connected vs. pipeline, also realize that even such solution paradigm using multi-agent collaboration/orchestration seem to overlap with recent work like OpenClaw, it may represent an important direction of future agentic AI approach, which justify the value of this work to a broader audience. Thus I change to weak accept.

**Key Questions For Authors:**

Q1. The system design includes a memory module in coordinator, however, the memory module has not been described clearly, and the effectiveness of memory module has not been studied.

Q2. One primary aspect of the proposed method trying to address is numerical problems nested inside an engineering problem. The correctness of solution to such type of problems boils down to the correctness of Math form transformation. The pipeline design might help but it is not clear by how much?

Q3. The cost analysis is primarily discussing cost in terms of tokens or monetary measurement. It might also be helpful to consider computation cost such as FLOPS.

**Limitations:**

No.

One limitation is that the performance of the proposed design would be primarily relying on the backbone LLM models.

I'm not aware of any negative social impact of this work.

**Strengths And Weaknesses:**

S1. This papers studies the problem of open-ended engineering with multi-agent, which hasn’t been well studied before.

S2. Compared with previous work, the solution has some novelty, i.e, besides using a fixed pipeline, this paper also proposes to use fully connected agents, which promotes coordination between agents.

S3. The experiments are relative comprehensive, showing some advantages of proposed methods against SOTA methods.

W1. This paper lacks technique depth. The major contribution of this paper is the EngiAgent (multi-agent) pipeline, which is essentially prompt-based method. No sophisticated method or algorithm has been proposed.

W2. Since the solution is purely LLM-based, the effectiveness of the proposed method would be largely depending on the capabilities of LLMs, rather than the pipeline design.

W3. The evaluation benchmark is self-constructed, which makes the evaluation potentially biased.

---

> ### Author Rebuttal · Authors · 2026-03-31
>
> Dear Reviewer Z7AS,
>
> Thank you for your helpful feedback and insightful suggestions. Our detailed responses are provided below.
>
> >**W1: "The paper lacks technical depth...EngiAgent is largely prompt-based...no substantial algorithmic contribution."**
>
> WA1: We respectfully disagree with this comment. Our work goes beyond a prompt-based pipeline by formulating engineering feasibility as a core objective, characterized by failure modes such as missing constraints, data inconsistency, and physical infeasibility. Based on this, EngiAgent introduces a coordination mechanism with dynamic routing and constraint-aware verification, forming a closed-loop error correction process.
>
> This leads to improved feasibility. Empirically, EngiAgent consistently outperforms prior methods across multiple models, achieving a 7× improvement over prior SOTA on average. Ablation results show that removing these components reduces feasibility under the same base model, indicating that gains are largely driven by coordination rather than prompting or model capability alone. Our contribution lies in feasibility-oriented problem analysis and coordination-level design, demonstrating that effective agent architectures are themselves a key form of contribution.
>
> >**W2: "Since the solution is LLM-based...effectiveness largely depends on LLM capability rather than pipeline design."**
>
> WA2: We agree that LLM capability affects performance, but the improvements cannot be attributed to the model alone. Notably, under the same LLM, **zero-shot feasibility is 0%** for both Gemini-2.5 Flash and DeepSeek-V3-671B, while EngiAgent achieves substantial gains, **showing that LLM capability alone is insufficient**. **Ablation results** further show that removing the coordination mechanism leads to clear feasibility drops under the same base model, indicating that improvements are largely driven by the pipeline design.
>
> >**W3: "The evaluation benchmark is self-constructed, which makes the evaluation potentially biased."**
>
> WA3: There is currently no public benchmark specifically designed to evaluate engineering feasibility, as prior work largely relies on text-based LLM-as-a-judge evaluation (see Section 6.3). Our benchmark is constructed to fill this gap rather than favor a particular method.
>
> To reduce bias, all tasks are adapted from published papers with fully disclosed sources and span multiple engineering domains (Appendix B). To further support generalizability, we also evaluate EngiAgent on an additional dataset (see Reviewer iL68, A1), where results are consistent with our original findings, suggesting that the gains are not limited to a specific benchmark.
>
> >**Q1: "The memory module in the coordinator is not clearly described...and its effectiveness is not evaluated."**
>
> QA1: We clarify that the memory module serves as a working memory for the coordinator (see Figure 3), supporting state tracking, error accumulation, and informed routing across iterations rather than acting as an independent component. It maintains intermediate problem-solving states and historical debugging information, enabling the system to avoid repeated errors and improve decision-making in subsequent steps. While not evaluated as an isolated module, it is tightly integrated into the coordination mechanism and contributes to iterative error correction and convergence; implementation details are available in our open-source code, and we will further clarify its design and role in the revision.
>
> >**Q2: "The proposed method addresses numerical problems...correctness boils down to mathematical formulation...the pipeline design might help, but it is unclear by how much?"**
>
> QA2: The key issue is not only correct mathematical formulation, but maintaining constraint completeness and consistency throughout the solving process. Infeasibility mainly arises from missing constraints, data inconsistency, and violations of physical laws, rather than formulation errors alone.
>
> Empirically, EngiAgent **achieves an average seven-fold improvement** in feasibility over prior SOTA methods. It also significantly improves the consistency between numerical and feasible solutions: the ratio of feasible to numerical solutions increases from **25% (zero-shot) to 97.14% (EngiAgent)** on GPT-4o. Ablation results further show that replacing the fully connected coordinator with a fixed pipeline leads to feasibility drops of over 10% on average, confirming that the gains come from coordinated feasibility enforcement rather than formulation alone.
>
> >**Q3: "The cost analysis focuses on tokens and monetary cost...it would be helpful to also consider computational cost such as FLOPs."**
>
> QA3: We agree that computational cost such as FLOPs is valuable, but it is not accessible when using closed-source LLM APIs. Prior work and our baselines typically report token usage, runtime, and monetary cost, which we follow to ensure fair comparison and reflect practical deployment cost.

---

> > ### Author Rebuttal · Reviewer_Z7AS · 2026-04-01
> >
> > Thanks for all the clarification. I think I understand this paper better now.
> >
> > The primary contribution is Fully Connected Coordinator and fill the Feasibility Gap.
> >
> > Also, by EngiBench, the authors provide a quantifiable way to measure "feasibility," which is hard to track historically.
> >
> > Duration cost seems significant but might be due to the complexity and API call.
> >
> > Such solution paradigm using multi-agent collaboration/orchestration seem to overlap with recent work like OpenClaw, but not exactly the same.
> >
> > Overall, I think this might still be a work that good to know by ICML audience, I will adjust my score accordingly.

---

> > > ### Author Response · Authors · 2026-04-03
> > >
> > > Dear Reviewer Z7AS,
> > >
> > > Thank you for your thoughtful feedback and for your positive assessment. We truly appreciate it.
> > >
> > > We are especially grateful for your recognition of our contributions and for noting that this work is valuable to the ICML audience.

---

### Official Review · Reviewer_S5Cn · 2026-03-12

**Soundness:** 3
**Presentation:** 3
**Significance:** 3
**Originality:** 3
**Overall Recommendation:** 4
**Confidence:** 4

**Summary:**

Many thanks to the authors for this submission. It was an enjoyable read. Here is my take on what you set out to accomplish. The objective of the proposed multi-agent LLM based framework named EngiAgent is to provide an open-ended engineering problem-solving system whose only criteria for success is that the solution is feasible, both numerically and with respect to all applicable data, physical, operational, and procedural constraints. The framework contains five individual agents, namely the analyzer, modeler, verifier, solver, and evaluator, as well as a fully connected coordinator and memory component to allow feedback routing and prevent pipeline failure modes. In addition to its role in routing feedback, the coordinator provides error identification, and therefore, it can also provide specific error resolution strategies; e.g., a forced agent switching strategy to avoid infinite loops of debugging. The authors also develop a benchmark set of 53 engineering problems drawn from four separate domains, and then they evaluate the feasibility of these problems through both executable numerical verification and human confirmation. The results demonstrate a significant improvement in the number of feasible solutions generated by the EngiAgent framework relative to modified agent-based models utilizing three different LLM models: GPT-4o, Gemini-2.5 Flash, and DeepSeek-V3-671B. The primary goal of this research, or at least based on my interpretation of it, is to address the disparity in the literature between "plausible-looking" solutions versus feasible solutions under the influence of real-world constraints. I think one of the most important issues that this paper tries to raise is that how do we properly design and evaluate models of LLMs that are capable of acting (agently), when the only measure of their success is not merely textual coherence?

**Compliance With Llm Reviewing Policy:**

Affirmed.

**Final Justification:**

The rebuttal reinforced my prior assessment.

**Key Questions For Authors:**

1.	Will EngiAgent work on an even broader range of different kinds of engineering problems (a much larger number of solvers, nonlinear constraint sets, stochastic constraint sets, messy table-based data)? These could influence what I would recommend.
2.	What are the actual routing/termination rules (or decision prompts) for the fully connected coordinator? And can you provide some type of analysis regarding failure in routing or looping in the coordinator?
3.	Are there significant differences in false positive/false negative rates between the verifier and human engineers? And how often will the verifier reject formulations that are functionally identical to the formulation being verified?

**Limitations:**

Yes. The paper includes an impact statement acknowledging risks of misuse/overreliance and frames EngiAgent as assistive with human oversight, while emphasizing feasibility verification and transparent intermediate outputs.

**Strengths And Weaknesses:**

Soundness

Strengths.
The paper provides clear definitions and explanations of the concept of feasibility and why text-only assessment is not appropriate in the context of engineering applications, with the helpful categorization of the different modes of failure in the process (vague modeling, modification of key data, violation of physical laws, over-constraint), and the example to support this. The design of the EngiAgent is clear and logical, with the different roles of analysis/modeling/verification/solving/evaluation fitting the process that is expected in engineering applications, and the Verifier’s emphasis on semantic/data/constraint consistency appropriate to the task of the feasibility study. The emphasis on the binary nature of the task, with the assessment by execution and the judgment of the expert, is a good methodological approach, given the thesis that the authors are presenting. The gains shown over the various adapted agents are large, and the ablations support the importance of the Verifier and the fully connected coordinator.

Weaknesses.
The number of tasks in the dataset is relatively low (53), and though the authors claim this is high-quality and expert-approved, the representativeness of the tasks to the full range of engineering applications is not clear, which reduces the confidence that the gains shown will be replicated outside the specific benchmark. The baselines are adapted from other applications (research automation, data science agents, math modeling agents), and though this is understandable, the thesis that the authors are presenting might be better served by more direct comparisons with other applications that are more relevant to the task, such as those in the optimization of engineering applications. The binary nature of the task, with the emphasis on the assessment of the feasibility, might not be sufficient, as the degrees to which the constraints are satisfied could be important, and the authors might have shown more diagnostics, such as the most frequently violated constraints, the margin to feasibility, etc. The decision process of the coordinator, though clear conceptually, might be better formalized to understand the effectiveness of the routing process, with the different types of error that might be encountered.

Presentation

Strengths.
The diagram for the workflow and the comparison of fixed-pipeline vs. fully connected are easy to understand and make the design intuitive. The paper is well structured, with motivation for feasibility, then failures, then architecture, then dataset, and finally experiments/ablations/cost.

Weaknesses.
There are some important reproducibility details that seem to be relegated to the appendices or left unspecified in the main paper (such as prompting, memory schema, solver backend selection, etc.). The paper has multiple evaluation metrics (IE/DR/MO/UH, along with feasibility), but it would be helpful to understand how these metrics are calculated, especially in relation to feasibility, since it was noted that using textual metrics can be misleading.

Significance

Strengths.
The focus on feasibility first evaluation is particularly relevant at this time as many agentic LLMs evaluate textual plausibility instead of feasibility when attempting to solve real world problems. The engineering domain is particularly well suited for evaluating solutions based on their ability to exist (be feasible) rather than simply being plausible. The dynamic routing of requests from one agent to another (the coordinator concept) has a high probability of having an impact on how "engineering grade" agentic models are built if it can also be shown to be reliable.

Weaknesses.
At this point in time, the scale and scope of the benchmark do not give us enough information to provide confidence that it will have significant impacts. More comprehensive engineering datasets with greater variety and larger scales would strengthen the significance of this work.

Originality
Strenghts.
The fully connected coordinator versus pipeline (dynamic feedback routing) is a useful systems-level contribution, particularly suited for identifying feasibility failure(s) at multiple points. The focus on feasibility as the main target for execution with verifiable results provides an interesting shift in perspective when compared to much of the existing work that evaluates LLM agents.

Weaknesses.
Agent functionality (analyze/model/verify/solve/evaluate), and reflection/retry have existed in previous agent architectures, the novel aspect of this work lies in the coordination connectivity and feasibility-centric approach, as opposed to the development of new learning algorithms or coordination theories.

---

> ### Author Rebuttal · Authors · 2026-03-31
>
> Dear Reviewer S5Cn,
>
> We thank the reviewer for the recognition of our work on feasibility-centered engineering reasoning and the design of EngiAgent, and will incorporate the suggestions in the revision.
>
> >**Q1: "Will EngiAgent work on a broader range of engineering problems..."**
>
> QA1: EngiAgent is a modular and solver-agnostic framework that can extend to a broad range of engineering problems, including different solvers, nonlinear or stochastic constraints, and heterogeneous data such as table-based inputs. Its coordination mechanism operates at the levels of analysis, modeling, and verification, and is not tied to specific solver types or constraint forms. As shown in Appendix G, the framework already supports diverse tools and problem settings, demonstrating its flexibility. We will clarify this in the revision.
>
> >**Q2: "What are the routing and termination rules...for the fully connected coordinator..."**
>
> QA2: The routing and termination rules of the coordinator are based on structured decision logic and prompting, with all decision prompts available in our anonymized open-source repository. Appendix E (Coordinator Routing Trace Examples) provides real routing traces with detailed records of trigger signals, reasoning, strategies, and outcomes, showing that the coordinator performs interpretable diagnostic reasoning rather than heuristic switching. Termination is explicitly controlled via a continue/stop signal. The traces also show how the system avoids ineffective loops through iterative debugging and state tracking, and converges to feasible solutions over multiple steps.
>
> >**Q3: "Are there differences in FP/FN rates between the verifier and human engineers..."**
>
> QA3: Our experiments across three models show that both false positive and false negative rates of the verifier are below 10%, indicating strong reliability. This is further supported by the feasible to numerical ratio, which exceeds 95% across all models, suggesting that most accepted solutions satisfy the constraints and that false positives are low.
>
> The verifier evaluates solutions based on semantic consistency, constraint completeness, and data consistency to ensure alignment with the original engineering problem. For functionally equivalent formulations, as long as they lead to equivalent constraint systems and feasible solutions, they are generally accepted.
>
> >**W1: "The dataset is relatively small...representativeness is unclear...limiting generalization." + weakness in significance**
>
> WA1: While the dataset contains 53 tasks, all are drawn from top-tier sources. We will expand it to improve coverage. Results on an additional dataset (see Reviewer iL68, A1) are consistent with our findings, suggesting the gains generalize beyond a specific benchmark.
>
> >**W2: "Baselines are from other domains...need more relevant engineering comparisons."**
>
> WA2: While domain-specific comparisons are valuable, there are no widely adopted baselines for open-ended engineering problem solving. Existing optimization agents focus on well-defined formulations and do not handle open-ended modeling, complex constraints, or simulator interactions in our setting. To strengthen the comparison, we additionally evaluate OpenClaw (see Reviewer ZxV3, WA2), and the results remain consistent, with EngiAgent achieving higher solution feasibility.
>
> >**W3: "The binary feasibility assessment...more diagnostics...would help."**
>
> WA3: Feasibility is defined by whether fundamental engineering constraints are satisfied; any violation makes a solution non-deployable, motivating a binary criterion. Appendix B.6–B.7 provide the annotation protocol and representative borderline cases.
>
> We agree that more detailed diagnostics are valuable. Due to space limitations, we report an analysis using DS-Agent with GPT-4o, where the feasible-to-numerical ratio is the lowest. We categorize infeasible solutions into four types: (1) violating physical laws (42.1%); (2) vague modeling (36.8%); (3) altering key data (15.8%); and (4) over-constraining the model (5.3%). More detailed diagnostics will be included in the revision.
>
> >**W4: "The coordinator decision process...could be better formalized..."**
>
> WA4: To clarify, the coordinator’s routing is dynamic and context-dependent, guided by LLM-based analysis of error signals, historical attempts, and agent statistics rather than fixed deterministic rules. In practice, error types are categorized to inform routing decisions, and historical statistics are used to guide retries. We will clarify this in the revision.
>
> >**W5 [weakness in originality] "Agent functionality..."**
>
> WA5: While these components exist in prior work, our contribution lies in their coordination for feasibility-oriented engineering reasoning. We introduce a fully connected coordination mechanism for dynamic routing and consistency enforcement, explicitly targeting engineering feasibility and addressing missing constraints and inconsistency not handled in prior frameworks.

---

> > ### Author Rebuttal · Reviewer_S5Cn · 2026-04-03
> >
> > The rebuttal is helpful and addresses several of my questions, especially regarding coordinator routing / termination behavior, verifier reliability, and the rationale for feasibility-first evaluation. I appreciate these clarifications, and they improve the presentation and scope framing of the work. That said, my concerns are still partially resolved rather than fully resolved. My main reservations remain about benchmark scale / representativeness, broader generalization across engineering settings, and the need for somewhat stronger formalization / diagnostics around the coordinator and feasibility evaluation. The rebuttal improves my understanding of the paper, but it does not materially change my overall assessment, and I am keeping my score unchanged.

---

> > > ### Author Response · Authors · 2026-04-07
> > >
> > > Dear Reviewer S5Cn,
> > >
> > > Thank you for your positive evaluation and for recognizing the value of our rebuttal. We appreciate your acknowledgment that our clarifications have improved the presentation and scope of the work. Due to rebuttal space limitations, we provide additional details below.
> > >
> > > >**Benchmark.**
> > >
> > > We place strong emphasis on benchmark quality, difficulty, and representativeness. Such evaluation problems are inherently scarce, as they require careful curation and expert validation. All problems in our benchmark are selected from top-tier journal and conference papers, and further constructed and verified by domain experts across multiple engineering fields to ensure both high quality and practical relevance. As detailed in Appendix B.1, B.2, and B.8, the dataset spans multiple engineering domains, maintains a balanced category distribution, and covers a wide range of problem complexities, from moderate cases with 5 to 20 constraints to highly complex instances with over 50 constraints or more than 100 parameters.
> > >
> > > We view this benchmark itself as a valuable resource and are actively working to further expand its scale and coverage. We agree that broader coverage would strengthen the evaluation, and this will be a key direction in future versions.
> > >
> > > >**EngiAgent Generalization.**
> > >
> > > EngiAgent is designed as a modular and solver-agnostic framework, where the coordination mechanism is centered on feasibility constraints rather than any specific solver or task setting. As shown in Appendix G, it supports diverse tools and problem settings without requiring changes to the core architecture. In addition, we evaluate EngiAgent on a newly constructed dataset (see response to Reviewer iL68, A1), where we observe consistent performance trends, providing further evidence of stable generalization across different settings.
> > >
> > > >**Formalization.**
> > >
> > > We also agree that stronger formalization and diagnostic tools are valuable. Formal analysis of agent systems is an important future direction, although mature methodologies for rigorous verification of such workflows remain limited. In this work, we instead focus on making the coordination process transparent and auditable through structured execution. As shown in Appendix E, representative execution traces illustrate how different failure types are identified, routed, and resolved. For example, code-level import errors are routed to the Modeler, as they are diagnosed as structural modeling issues rather than conceptual or optimization-stage errors. For feasibility evaluation, we make the feasibility criterion explicit in Section 5 and Appendix B.5, and further support it with a structured annotation protocol and clearly defined standards, with boundary-case examples provided in Appendix B.6 and B.7.

---

### Official Review · Reviewer_ZxV3 · 2026-03-13

**Soundness:** 3
**Presentation:** 2
**Significance:** 2
**Originality:** 2
**Overall Recommendation:** 4
**Confidence:** 4

**Summary:**

This paper addresses the critical gap in applying large language models (LLMs) to engineering problem-solving, where existing methods often fail to ensure solution feasibility under physical, operational, and data constraints. It introduces EngiAgent, a multi-agent framework with a fully connected coordinator that simulates expert workflows by dividing tasks into problem analysis, modeling, verification, solving, and evaluation, enabling dynamic feedback routing to correct feasibility issues at each stage. The contributions include identifying feasibility as a core engineering requirement, constructing a specialized dataset for evaluation, and demonstrating that EngiAgent significantly outperforms prior methods in feasibility across diverse engineering tasks and multiple LLMs.

**Compliance With Llm Reviewing Policy:**

Affirmed.

**Final Justification:**

The author has answered my questions, especially in the case where the success rate of openclaw was higher. This proves that the author has indeed conducted specific thinking and experiments.

**Key Questions For Authors:**

The open-source framework OpenClaw, when paired with anthropic/claude-sonnet-4.6, has already achieved a task execution accuracy/credibility rate of 86.9%. However, the authors claim that the feasibility of current models in solving engineering problems is merely around 13%. Please provide a clear explanation for this discrepancy, and ideally demonstrate the advantages of your proposed method through a direct comparison.

**Limitations:**

The issue of large model feasibility sounds interesting, but the authors only used simple examples for testing, which undermines the credibility of the results. It is recommended to adopt more complex examples that are closer to real-world engineering scenarios. The authors failed to describe the consequences that would arise when the proposed method is applied but generates incorrect solutions, as it is clear that the feasibility has not yet reached the level of practical usability expected by humans.

**Strengths And Weaknesses:**

Strength：
The authors clearly regarded feasibility as the core indicator for engineering modeling, and established a dedicated evaluation dataset and quantification system.
The method proposed in this paper improves the strategic correctness of large models to a certain extent, achieving state-of-the-art (SOTA) performance within constrained scopes across multiple model and task combinations.
Weakness：
The introduction and main text contain excessive verbose, non-substantive content lacking proper literature citations; we recommend streamlining the language to enhance conciseness and rigor.
The solution feasibility of the proposed method does not surpass that of mainstream open-source agent frameworks available publicly, such as OpenClaw paired with Claude-Sonnet-4.6.
The architecture of EngiAgent is highly similar to existing mainstream agent models, with no evident high-level novelty; its problem-solving approach is application-oriented, offering limited scientific contribution.

---

> ### Author Rebuttal · Authors · 2026-03-31
>
> Dear Reviewer ZxV3,
>
> Thank you for the insightful comments. We provide our detailed responses below.
>
> >**W1: "The introduction and main text are overly verbose and lack proper citations..."**
>
> WA1: We will revise the manuscript to improve clarity and conciseness, and further strengthen citations. The current version already covers key related work, and we welcome suggestions for additional relevant references.
>
> >**W2: "The solution feasibility...does not exceed that of mainstream frameworks, such as OpenClaw paired with Claude-Sonnet-4.6."**
>
> WA2: We include OpenClaw as a baseline in additional experiments under the same setting. We use GPT-5.4 as the base model, as it outperforms the Claude-Sonnet-4.6 mentioned by the reviewer on PinchBench and allows a more consistent comparison with EngiAgent evaluated on GPT-series models. The results are:
>
> | Method     | Num.   | Feas.  |
> |------------|--------|--------|
> | Zero-shot  | 22.64% | 5.66%  |
> | OpenClaw   | 86.79% | 56.60% |
> | EngiAgent  | 66.04% | **64.15%** |
>
> This result highlights our key finding: high numerical success does not imply engineering feasibility. OpenClaw relies on code execution and iterative debugging to obtain solutions, but under complex constraints it often omits or implicitly relaxes critical constraints, leading to numerically valid yet infeasible outcomes (e.g., violating power balance or capacity limits). In contrast, EngiAgent enforces constraint completeness and consistency through coordinated verification, resulting in a higher proportion of feasible solutions, showing that execution and solver capabilities alone are insufficient without explicit constraint validation mechanisms.
>
> OpenClaw was not included in the original submission because it is concurrent work and had not undergone peer review. Specifically, Clawdbot was released on Nov 24, 2025 and renamed to OpenClaw on Jan 29, 2026.
>
> >**W3: "EngiAgent is similar to existing agent models...with limited novelty and scientific contribution."**
>
> WA3: Our contribution is not merely the introduction of a multi-agent architecture, but that the architecture is **systematically designed based on the engineering problem-solving workflow (analysis–modeling–verification–solving)**, rather than a direct reuse of existing mainstream approaches. We **focus on the long-overlooked issue of feasibility in engineering**, and propose a clear problem formulation, an evaluation framework, and a coordination mechanism that explicitly ensures engineering feasibility. Therefore, our work is not merely application-oriented; it provides a new perspective at the levels of problem formulation, evaluation criteria, and solution mechanisms, addressing a critical gap that has been largely overlooked and thus constituting a meaningful scientific contribution.
>
> >**Q1: "The open-source framework OpenClaw, when paired with anthropic/claude-sonnet-4.6, reports ~86.9% execution success...whereas the authors report ~13% feasibility...please clarify the discrepancy and provide direct comparison."**
>
> QA1: We are not certain which benchmark the reported 86.9% corresponds to, but the value appears close to execution success rates reported on PinchBench, which measure task execution ability rather than strict engineering feasibility. As shown in WA2, OpenClaw indeed achieves a high numerical success rate, but its proportion of feasible solutions remains clearly lower than that of EngiAgent. This indicates that execution success and engineering feasibility are fundamentally different metrics, which is exactly the distinction our work aims to address.
>
> >**L1: "The issue of large model feasibility sounds interesting, but ... simple examples for testing...undermines credibility...more complex, real-world engineering scenarios are recommended."**
>
> LA1: The problems in our dataset are derived from published top-tier journal or conference papers across multiple engineering domains, preserving key constraints and structure, and thus are not simplified examples. In Appendix B.8, we further report the distribution of constraints and parameters, showing a broad range of complexity, including high-complexity cases. This supports that our benchmark reflects realistic and diverse engineering scenarios.
>
> >**L2: "The authors did not discuss the consequences...when incorrect solutions are generated...feasibility remains below practical usability."**
>
> LA2: From our results, EngiAgent achieves a high ratio of feasible solutions among generated numerical solutions (97.14%, 96.42%, and 95.23% for GPT, Gemini, and DeepSeek, respectively), which is consistently higher than all baselines and is not low by standards in many engineering domains. While errors may still occur, our goal is to improve reliability rather than replace human experts. As noted in our impact statement, we view the system as an assistive tool, and human oversight remains essential. We will clarify this in the revision.

---

> > ### Author Rebuttal · Reviewer_ZxV3 · 2026-04-02
> >
> > The author has answered my questions, especially in the case where the success rate of openclaw was higher. This proves that the author has indeed conducted specific thinking and experiments.

---

> > > ### Author Response · Authors · 2026-04-03
> > >
> > > Dear Reviewer ZxV3,
> > >
> > > Thank you for your thoughtful feedback and for your positive acknowledgment. We truly appreciate it.
> > >
> > > We are grateful for your recognition that your concerns have been fully addressed and for your appreciation of our analysis and experiments. We also appreciate the opportunity to highlight the strengths of EngiAgent compared to OpenClaw, particularly in improved feasibility.

---

### Official Review · Reviewer_iL68 · 2026-03-18

**Soundness:** 3
**Presentation:** 4
**Significance:** 3
**Originality:** 3
**Overall Recommendation:** 5
**Confidence:** 3

**Summary:**

This paper proposes EngiAgent, a multi-agent system for solving open-ended engineering problems with an emphasis on feasible solutions rather than only correct formulations or executable code. The system uses five specialized agents—Analyzer, Modeler, Verifier, Solver, and Evaluator—coordinated by a fully connected controller that routes feedback dynamically across stages. The paper reports substantial gains in feasibility across four engineering domains and compares the fully connected coordinator against a fixed pipeline baseline.

**Compliance With Llm Reviewing Policy:**

Affirmed.

**Final Justification:**

Open-ended Engineering Problems mean a lot in Real-World Scenarios while very underestimated by LLM Application Researchers these days.
I believe that the paper is a good start in this domain.

**Key Questions For Authors:**

How well do the results generalize beyond the 53-problem benchmark?
Can the authors more cleanly separate the contribution of the fully connected coordinator from verifier and prompt-engineering effects?
How well would the framework transfer to softer engineering design tasks without a crisp feasibility check?
Do the authors expect the same gains for larger-scale engineering optimization problems?

**Limitations:**

yes

**Strengths And Weaknesses:**

This is one of the strongest papers in the set. The problem formulation is very good: the paper correctly points out that in engineering, feasibility is often more important than elegance, fluency, or even partial correctness, and this distinction is genuinely important. The architecture is clear and practically motivated, and I also appreciate that the paper compares the fully connected coordinator to a fixed pipeline and reports concrete feasibility improvements with relatively similar or even lower cost. The ablation evidence is helpful and makes the paper feel more mature than many other multi-agent system papers.

The main weakness is that the empirical scope is still somewhat limited relative to the ambition of the framing. The dataset contains 53 high-quality problems across four engineering domains, which is a useful start, but still small for supporting very broad claims about open-ended engineering problem solving in general. In addition, although the fully connected coordinator is presented as the key idea, the ablation study also shows that prompt engineering, the verifier, and forced switching all matter substantially. This means the gain is real, but it is not solely attributable to the coordinator itself; the full system is a fairly composite design. Finally, the paper is strongest in settings where feasibility can be checked relatively concretely through modeling, verification, and solver outcomes. That is a strength for the current evaluation, but it leaves open how well the framework will transfer to engineering settings where feasibility is softer, more qualitative, or multi-objective.

---

> ### Author Rebuttal · Authors · 2026-03-31
>
> Dear Reviewer iL68,
>
> Thank you for the encouraging comment that “This is one of the strongest papers in the set”. We sincerely appreciate this recognition and thank the reviewer for all the valuable suggestions, which we will carefully incorporate into the revision.
>
> >**Q1: "How well do the results generalize beyond the 53-problem benchmark?"**
>
> A1: We thank the reviewer for the suggestion on evaluating generalization beyond the 53-problem benchmark. In the revised version, we extend our evaluation to an additional dataset, EngiBench [1], which is constructed from published research papers, EngiBench consists of engineering modeling tasks from competition settings.
>
> Since the core objective of EngiAgent is to produce feasible numerical solutions, we select 19 tasks from EngiBench that involve constraint-based modeling with clearly defined feasibility criteria, and exclude purely analytical or unconstrained problems. Following our dataset construction process, we further add explicit feasibility constraints and verification criteria for each task.
>
> The results are summarized below:
>
> Table 1. Numerical and Feasibility Rates.
>
> | Dataset           | Num. ↑ | Feas. ↑ |
> |------------------|--------|---------|
> | Original Dataset | 66.04% | 64.15%  |
> | EngiBench        | 63.16% | 52.63%  |
>
> Table 2. Total Performance.
>
> | Dataset           | IE ↑ | DR ↑ | MO ↑ | UH ↑ | Avg. ↑ |
> |------------------|------|------|------|------|--------|
> | Original Dataset | 8.67 | 7.74 | 7.05 | 7.41 | 7.72   |
> | EngiBench        | 7.78 | 6.36 | 6.14 | 6.20 | 6.62   |
>
> Table 3. Feasible-Only Performance.
>
> | Dataset           | IE ↑ | DR ↑ | MO ↑ | UH ↑ | Avg. ↑ |
> |------------------|------|------|------|------|--------|
> | Original Dataset | 8.86 | 8.13 | 7.53 | 7.94 | 8.12   |
> | EngiBench        | 7.50 | 6.72 | 7.96 | 7.50 | 7.42   |
>
> Overall, EngiAgent maintains comparable performance on EngiBench with consistent trends across metrics, indicating that the method generalizes beyond the original benchmark. Moreover, as EngiBench tasks are derived from real-world engineering modeling competitions, these results further support the applicability of our approach in practical and diverse scenarios.
>
> [1] Zhou, X., Wang, X., He, Y., Wu, Y., Zou, R., Cheng, Y., ... & Zhao, J. (2025). Engibench: A benchmark for evaluating large language models on engineering problem solving. arXiv preprint arXiv:2509.17677.
>
> >**Q2: "Can the authors more cleanly separate the contribution of the fully connected coordinator from verifier and prompt-engineering effects?"**
>
> A2: We thank the reviewer for the insightful comment. We agree that the improvements are not attributable to a single component, but arise from the interaction of multiple modules. As shown in Section 6.5 (Ablation Study), removing any component, including prompt engineering, verifier, or coordinator, leads to clear performance degradation, indicating that the system is inherently compositional.
>
> In this context, the fully connected coordinator is not an isolated source of gains, but the mechanism that enables structured interaction across modules through dynamic routing and global state coordination. While components such as the verifier and prompt design improve local correctness, the coordinator integrates them into a consistent solving process. We will further clarify this distinction and more clearly separate the roles of different components in the revision.
>
>
> >**Q3: "How well would the framework transfer to softer engineering design tasks without a crisp feasibility check?"**
>
> A3: Our framework can be transferred to softer engineering design tasks, where the key lies in extending the design of the verifier. Specifically, the original crisp feasibility check can be generalized to a more flexible evaluation mechanism, making it applicable to scenarios without a clearly defined feasibility criterion. We will further explore this direction and clarify the scope of our method in the revision, as well as discuss its potential extensions to softer engineering tasks.
>
>
> >**Q4: "Do the authors expect the same gains for larger-scale engineering optimization problems?"**
>
> A4: We expect the gains to transfer to larger-scale engineering optimization problems. As shown in Appendix B.8 (Complexity Distribution of Engineering Tasks), our dataset covers a wide range of problem difficulties, and since the tasks are constructed from published papers, they naturally reflect the complexity of real-world engineering problems. Empirically, we observe consistent performance across different complexity levels. For example, among the ten most constraint-intensive tasks, EngiAgent produces feasible solutions for five cases, which is comparable to its feasibility rate on the full set. This suggests that the effectiveness of the framework remains stable as problem complexity increases.

---

> > ### Author Rebuttal · Reviewer_iL68 · 2026-04-03
> >
> > Issue Resolved

---

> > > ### Author Response · Authors · 2026-04-03
> > >
> > > Dear Reviewer iL68,
> > >
> > > Thank you for your encouraging feedback and for your continued positive support. We truly appreciate it.
> > >
> > > We are grateful for your recognition that our response has fully addressed your concerns.

---

### Decision · Program_Chairs · 2026-04-30

**Decision:**

Accept (regular)

**Comment:**

The rebuttal stimulated some discussion and helped clarify several of the reviewers main concerns. Overall, the reviewer consensus is positive: while concerns remain about benchmark scale and the degree of methodological novelty, the paper identifies an important problem and demonstrates meaningful empirical gains in feasibility-oriented engineering problem solving. I believe the program would benefit from this paper being presented.